# MOAT: Alternating Mobile Convolution and Attention Brings Strong Vision Models

**Chenglin Yang**[1][*]**, Siyuan Qiao**[2]**, Qihang Yu**[1]**, Xiaoding Yuan**[1]**, Yukun Zhu**[2]**,**
**Alan Yuille**[1]**, Hartwig Adam**[2]**, Liang-Chieh Chen**[2]
[1]The Johns Hopkins University
[2]Google Research

## Abstract

This paper presents MOAT, a family of neural networks that build on top of **MO**bile convolution (*i.e.*, inverted residual blocks) and **AT**tention. Unlike the current works that stack separate mobile convolution and transformer blocks, we effectively merge them into a MOAT block. Starting with a standard Transformer block, we replace its multi-layer perceptron with a mobile convolution block, and further reorder it before the self-attention operation. The mobile convolution block not only enhances the network representation capacity, but also produces better downsampled features. Our conceptually simple MOAT networks are surprisingly effective, achieving 89.1% / 81.5% top-1 accuracy on ImageNet-1K / ImageNet-1K-V2 with ImageNet-22K pretraining. Additionally, MOAT can be seamlessly applied to downstream tasks that require large resolution inputs by simply converting the global attention to window attention. Thanks to the mobile convolution that effectively exchanges local information between pixels (and thus cross-windows), MOAT does not need the extra window-shifting mechanism. As a result, on COCO object detection, MOAT achieves 59.2% AP$^{\text{box}}$ with 227M model parameters (single-scale inference, and hard NMS), and on ADE20K semantic segmentation, MOAT attains 57.6% mIoU with 496M model parameters (single-scale inference). Finally, the tiny-MOAT family, obtained by simply reducing the channel sizes, also surprisingly outperforms several mobile-specific transformer-based models on ImageNet. The tiny-MOAT family is also benchmarked on downstream tasks, serving as a baseline for the community. We hope our simple yet effective MOAT will inspire more seamless integration of convolution and self-attention. Code is publicly available.[1]

## 1 Introduction

The vision community has witnessed the prevalence of self-attention (Bahdanau et al., 2015) and Transformers (Vaswani et al., 2017). The success of Transformers in natural language processing motivates the creation of their variants for vision recognition. The Vision Transformer (ViT) (Dosovitskiy et al., 2021) has great representation capacity with global receptive field. However, it requires pretraining on a large-scale proprietary dataset (Sun et al., 2017). Its unsatisfying performance, when trained with a small number of images, calls for the need of better training recipes (Touvron et al., 2021a; Steiner et al., 2021) or architectural designs (Liu et al., 2021; Graham et al., 2021). On the other hand, ConvNet has been the dominant network choice since the advent of AlexNet (Krizhevsky et al., 2012) in 2012. Vision researchers have condensed the years of network design experience into multiple principles, and have started to incorporate them to vision transformers. For example, there are some works adopting the ConvNet's hierarchical structure to extract multi-scale features for vision transformers (Liu et al., 2021; Fan et al., 2021; Wang et al., 2022), and others proposing to integrate the translation equivariance of convolution into transformers (Graham et al., 2021; d'Ascoli et al., 2021; Xiao et al., 2021).

Along the same direction of combining the best from Transformers and ConvNets, CoAtNet (Dai et al., 2021) and MobileViT (Mehta & Rastegari, 2022a) demonstrate outstanding performance by

---

[*]Work done while an intern at Google.
[1]Official code in TensorFlow: https://github.com/google-research/deeplab2

stacking Mobile Convolution (MBConv) blocks (*i.e.*, inverted residual blocks (Sandler et al., 2018)) and Transformer blocks (*i.e.*, a self-attention layer and a Multi-Layer Perceptron (MLP)). However, both works focus on the macro-level network design. They consider MBConv and Transformer blocks as individual separate ones, and systematically study the effect of stacking them to strike a better balance between the remarkable efficiency of MBConv and strong capacity of Transformer.

In this work, on the contrary, we study the *micro-level* building block design by taking a deeper look at the combination of MBConv and Transformer blocks. We make two key observations after a careful examination of those blocks. First, the MLP module in Transformer block is similar to MBConv, as both adopt the inverted bottleneck design. However, MBConv is a more powerful operation by employing one extra $3 \times 3$ depthwise convolution (to encode local interaction between pixels), and more activation (Hendrycks & Gimpel, 2016) and normalization (Ioffe & Szegedy, 2015) are employed between convolutions. Second, to extract multi-scale features using Transformer blocks, one may apply the average-pooling (with stride 2) to input features before the self-attention layer. However, the pooling operation reduces the representation capacity of self-attention. Our observations motivate us to propose a novel **MO**bile convolution with **AT**tention (MOAT) block, which efficiently combines MBConv and Transformer blocks. The proposed MOAT block modifies the Transformer block by first replacing its MLP with a MBConv block, and then reversing the order of attention and MBConv. The replacement of MLP with MBConv brings more representation capacity to the network, and reversing the order (MBConv comes before self-attention) delegates the downsampling duty to the strided depthwise convolution within the MBConv, learning a better downsampling kernel.

We further develop a family of MOAT models by stacking and increasing the channels of network blocks. Surprisingly, our extremely *simple* design results in a remarkable impact. On the challenging ImageNet-1K classification benchmark (Russakovsky et al., 2015), our model (190M parameters) achieves 86.7% top-1 accuracy without extra data. When further pretraining on ImageNet-22K, our best model (483M parameters) attains 89.1% / 81.5% top-1 accuracy on ImageNet-1K (Tab. 2) / ImageNet-1K-V2 (Tab. 9), setting a new state-of-the-art.

Additionally, MOAT can be *seamlessly* deployed to downstream tasks that require large resolution inputs by simply converting the global attention to non-overlapping local window attention. Thanks to the MBConv that effectively exchanges local information between pixels (enabling cross-window propagation), MOAT does not need the extra window-shifting mechanism (Liu et al., 2021). As a result, on COCO object detection (Lin et al., 2014) and ADE20K semantic segmentation (Zhou et al., 2019), MOAT shows superior performances. Specifically, on COCO object detection (Tab. 3), our best model (227M parameters), achieves 59.2% AP$^{box}$ with single-scale inference and hard NMS, setting a new state-of-the-art in the regime of model size 200M with Cascade Mask R-CNN (Cai & Vasconcelos, 2018; He et al., 2017). On ADE20K semantic segmentation (Tab. 4), our best model (496M parameters), adopting DeepLabv3+ (Chen et al., 2018), attains 57.6% mIoU with single-scale inference, also setting a new state-of-the-art in the regime of models using input size $641 \times 641$.

Finally, to explore the scalability of MOAT models, we *simply* scale down the models by reducing the channel sizes (without any other change), resulting in the tiny-MOAT family, which also surprisingly outperforms mobile-specific transformer-based models, such as Mobile-Former (Chen et al., 2022c) and MobileViTs (Mehta & Rastegari, 2022a;b). Specifically, in the regime of model parameters 5M, 10M, and 20M, our tiny MOAT outperforms the concurrent MobileViTv2 (Mehta & Rastegari, 2022b) by 1.1%, 1.3%, and 2.0% top-1 accuracy on ImageNet-1K classification benchmark (Tab. 5). Furthermore, we benchmark tiny-MOAT on COCO object detection and ADE20K semantic segmentation.

In summary, our method advocates the design principle of simplicity. Without inventing extra complicated operations, the proposed MOAT block effectively merges the strengths of both mobile convolution and self-attention into one block by a careful redesign. Despite its conceptual simplicity, impressive results have been obtained on multiple core vision recognition tasks. We hope our study will inspire future research on seamless integration of convolution and self-attention.

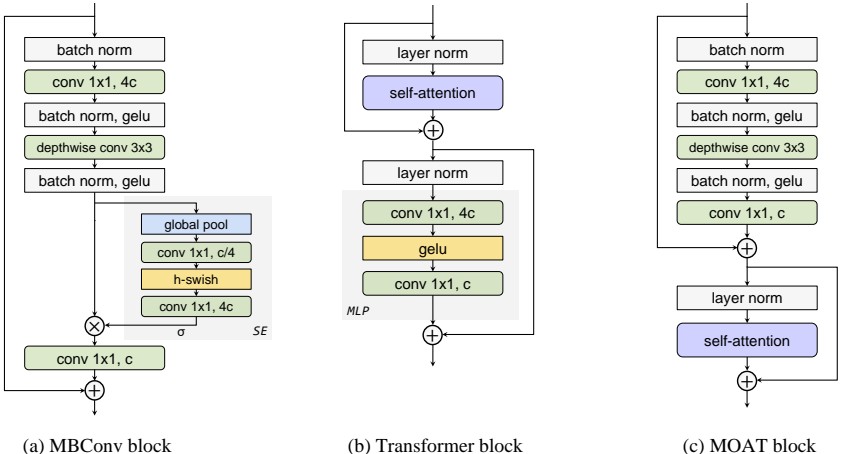

|  (a) MBConv block | (b) Transformer block | (c) MOAT block |

Figure 1: **Block comparison.** (a) The MBConv block (Sandler et al., 2018) employs the inverted bottleneck design with depthwise convolution and squeeze-and-excitation (Hu et al., 2018) applied to the expanded features. (b) The Transformer block (Vaswani et al., 2017) consists of a self-attention module and a MLP module. (c) The proposed MOAT block effectively combines them. The illustration assumes the input tensor has channels $c$.

## 2 METHOD

Herein, we review the Mobile Convolution (MBConv) (Sandler et al., 2018) and Transformer (Vaswani et al., 2017) blocks before introducing the proposed MOAT block. We then present MOAT, a family of neural networks, targeting at different trade-offs between accuracy and model complexity.

### 2.1 MOBILE CONVOLUTION AND TRANSFORMER BLOCKS

**MBConv block.** Also known as the inverted residual block, the Mobile Convolution (MBConv) (Sandler et al., 2018) block (Fig. 1 (a)) is an effective building block that has been widely used in mobile models (Howard et al., 2019; Mehta & Rastegari, 2022a) or efficient models (Tan & Le, 2019; Dai et al., 2021). Unlike the bottleneck block in ResNet (He et al., 2016a), the MBConv block employs the design of an "inverted bottleneck", together with the efficient depthwise convolution (Howard et al., 2017). Specifically, a $1 \times 1$ convolution is first applied to expand the input channels by a factor of 4. Then, a $3 \times 3$ depthwise convolution is used to effectively capture the local spatial interactions between pixels. Finally, the features are projected back to the original channel size via a $1 \times 1$ convolution, enabling a residual connection (He et al., 2016a). An optional Squeeze-and-Excitation (SE) (Hu et al., 2018) module (which uses the global information to re-weight the channel activation) may also be used after the depthwise convolution, following MobileNetV3 (Howard et al., 2019). Note that one could tune the channel expansion ratio and depthwise convolution kernel size for better performance. We fix them throughout the experiments for simplicity.

Formally, given an input tensor $x \in \mathbb{R}^{H \times W \times C}$ ($H, W, C$ are its height, width, and channels), the MBConv block is represented as follows:

$$\textbf{MBConv}(x) = x + (\mathcal{N}_2 \circ \mathcal{S} \circ \mathcal{D} \circ \mathcal{N}_1)(\text{BN}(x)), \qquad (1)$$

$$\mathcal{N}_1(x) = \text{GeLU}(\text{BN}(\textbf{Conv}(x))), \qquad (2)$$

$$\mathcal{D}(x) = \text{GeLU}(\text{BN}(\textbf{DepthConv}(x))), \qquad (3)$$

$$\mathcal{S}(x) = \sigma(\text{MLP}(\text{GAP}(x))) \cdot x, \qquad (4)$$

$$\mathcal{N}_2(x) = \textbf{Conv}(x), \qquad (5)$$

where BN, GeLU, GAP, and MLP stand for Batch Normalization (Ioffe & Szegedy, 2015), Gaussian error Linear Unit (Hendrycks & Gimpel, 2016), Global Average Pooling, and Multi-Layer Perceptron (with reduction ratio 4 and hard-swish (Ramachandran et al., 2017)), respectively. The MBConv block consists of four main functions: $\mathcal{N}_1$, $\mathcal{D}$, $\mathcal{S}$, and $\mathcal{N}_2$, which correspond to the $1 \times 1$ convolution for channel expansion (by $4\times$), $3 \times 3$ depthwise convolution, squeeze-and-excitation (Hu et al., 2018) ($\sigma$ is the sigmoid function), and $1 \times 1$ convolution for channel projection (by $4\times$), respectively.

**Transformer block.** The Transformer (Vaswani et al., 2017) block (Fig. 1 (b)) is a powerful building block that effectively captures the global information via the data-dependent self-attention operation. It consists of two main operations: self-attention and MLP. The self-attention operation computes the attention map based on the pairwise similarity between every pair of pixels in the input tensor, thus enabling the model's receptive field to encompass the entire spatial domain. Additionally, the attention map dynamically depends on the input, enlarging the model's representation capacity (unlike the convolution kernels, which are data-independent). The MLP operation contains two $1 \times 1$ convolutions, where the first one expands the channels (by $4\times$), the second one shrinks back the channels, and GeLU non-linearity is used in-between.

Formally, given an input tensor $x \in \mathbb{R}^{H \times W \times C}$, the Transformer block is represented as follows:

$$\textbf{Transformer}(x) = x + (\mathcal{M}_2 \circ \mathcal{M}_1 \circ \textbf{Attn})(\text{LN}(x)), \tag{6}$$

$$\mathcal{M}_1(x) = \text{GeLU}(\textbf{Conv}(\text{LN}(x))), \tag{7}$$

$$\mathcal{M}_2(x) = \textbf{Conv}(x), \tag{8}$$

where LN and Attn denote the Layer Normalization (Ba et al., 2016), and self-attention (Vaswani et al., 2017). The self-attention operation also includes a residual connection (He et al., 2016a), which is not shown in the equations for simplicity, while the MLP operation is represented by two functions $\mathcal{M}_1$ and $\mathcal{M}_2$, which correspond to the $1 \times 1$ convolution for channel expansion (by $4\times$) and $1 \times 1$ convolution for channel projection, respectively.

## 2.2 MOBILE CONVOLUTION WITH ATTENTION (MOAT) BLOCK

**Comparing MBConv and Transformer blocks.** Before getting into the architecture of our MOAT block, it is worthwhile to compare the MBConv (Sandler et al., 2018) and Transformer (Vaswani et al., 2017) blocks, which helps to understand our design motivations. Specifically, we make the following key observations.

First, both MBConv and Transformer blocks advocate the "inverted bottleneck" design, where the channels of input tensors are expanded and then projected by $1 \times 1$ convolutions. However, MBConv additionally employs a $3 \times 3$ depthwise convolution between those two $1 \times 1$ convolutions, and there are both batch normalization and GeLU activation between the convolutions.

Second, to capture the global information, the MBConv block may employ a Squeeze-and-Excitation (SE) module, while the Transformer block adopts the self-attention operation. Note that the SE module squeezes the spatial information via a global average pooling, while the self-attention module maintains the tensor's spatial resolution.

Third, the downsampling operation is performed at different places within the block. To downsample the features, the standard MBConv block uses the strided depthwise convolution, while the Transformer block, deployed in the modern hybrid model CoAtNet (Dai et al., 2021), adopts an average-pooling operation before the self-attention.

**MOAT block.** Given the above observations, we now attempt to design a new block that effectively merges the best from both MBConv and Transformer blocks. We begin with the powerful Transformer block, and gradually refine over it.

Based on the first observation, both MBConv and Transformer blocks employ the "inverted bottleneck" design. Since depthwise convolution could effectively encode local interaction between pixels, which is crucial for modeling the translation equivariance in ConvNets, we thus start to add the depthwise convolution to Transformer's MLP module. However, we did not observe any performance improvement until we also added the extra normalization and activations between convolutions.

For the second observation, we simply do not add the SE module to the MBConv block. The self-attention operation is kept to capture the global information.

We found the third observation critical. The downsampling operation (average-pooling) right before the self-attention operation in Transformer block slightly reduces its representation capacity. On the other hand, the MBConv block is well-designed for the downsampling operation with the strided depthwise convolution, which effectively learns the downsampling convolution kernel for each input channel. Therefore, we further reorder the "inverted bottleneck" (containing depthwise convolution) before the self-attention operation, delegating the downsampling operation to depthwise convolution.

In this way, we need no extra downsampling layer like average-pooling in CoAtNet (Dai et al., 2021), or patch-embedding layers in Swin (Liu et al., 2021) and ConvNeXt (Liu et al., 2022b). Finally, it results in our **MO**bile convolution with **AT**tention (MOAT) block, as illustrated in Fig. 1 (c).

Formally, given an input tensor $x \in \mathbb{R}^{H \times W \times C}$, the MOAT block is represented as follows:

$$\textbf{MOAT}(x) = x + (\textbf{Attn} \circ \mathcal{N}_2 \circ \mathcal{D} \circ \mathcal{N}_1)(\text{BN}(x)), \tag{9}$$

where MBConv (w/o SE) contains functions $\mathcal{N}_1$ (Eq. 2), $\mathcal{D}$ (Eq. 3), and $\mathcal{N}_2$ (Eq. 5), and Attn denotes the self-attention operation. The MOAT block then simply consists of MBConv (w/o SE) and the self-attention operation, successfully combining the best from the MBConv block and Transformer block into one (which we will show empirically).

## 2.3 META ARCHITECTURE

**Macro-level network design.** After developing the MOAT block, we then study how to effectively stack them to form our base model. We adopt the same strategy as the existing works (Liu et al., 2021; Wang et al., 2021b; Graham et al., 2021; Xiao et al., 2021; Dai et al., 2021; Mehta & Rastegari, 2022a). Specifically, we summarize several key findings from those works, and use them as design principles of our meta architecture.

- Employing convolutions in the early stages improves the performance and training convergence of Transformer models (Wu et al., 2021; Graham et al., 2021; Xiao et al., 2021).

- The Mobile Convolution (MBConv) (Sandler et al., 2018) blocks are also effective building blocks in the hybrid Conv-Transformer models (Dai et al., 2021; Mehta & Rastegari, 2022a).

- Extracting multi-scale backbone features benefits the downstream tasks, such as detection and segmentation (Liu et al., 2021; Wang et al., 2021b; Fan et al., 2021; Heo et al., 2021).

As a result, our meta architecture consists of the convolutional stem, MBConv blocks, and MOAT blocks. Additionally, through the ablation study in the appendix, we found the layer layout proposed by CoAtNet-1 (Dai et al., 2021) effective. We thus follow their layer layout, resulting in our base model MOAT-1. To form the MOAT model family, we then scale down or up MOAT-1 in the dimensions of number of blocks and number of channels, as shown in Tab. 1. We only scale the number of blocks in the third and fourth stages (out of five stages). The downsampling operation is performed in the first block of each stage. Note that our base model MOAT-1 and CoAtNet-1 share the same layer layout and channel sizes. However, we take a different scaling strategy: our MOAT is scaled up (or down) by alternatively increasing the depth and expanding the width between variants.

Table 1: **MOAT variants** differ in the number of blocks B and number of channels C in each stage.

| block | stride | MOAT-0 B | MOAT-0 C | MOAT-1 B | MOAT-1 C | MOAT-2 B | MOAT-2 C | MOAT-3 B | MOAT-3 C | MOAT-4 B | MOAT-4 C | tiny-MOAT-{0,1,2,3} B | tiny-MOAT-{0,1,2,3} C | | | |
|-------|--------|---|----|----|----|----|-----|----|------|----|------|---|-----|-----|-----|-----|
| conv | 2 | 2 | 64 | 2 | 64 | 2 | 128 | 2 | 160 | 2 | 256 | 2 | 32 | 40 | 56 | 80 |
| MBConv | 4 | 2 | 96 | 2 | 96 | 2 | 128 | 2 | 160 | 2 | 256 | 2 | 32 | 40 | 56 | 80 |
| MBConv | 8 | 3 | 192 | 6 | 192 | 6 | 256 | 12 | 320 | 12 | 512 | 3 | 64 | 80 | 112 | 160 |
| MOAT | 16 | 7 | 384 | 14 | 384 | 14 | 512 | 28 | 640 | 28 | 1024 | 7 | 128 | 160 | 224 | 320 |
| MOAT | 32 | 2 | 768 | 2 | 768 | 2 | 1024 | 2 | 1280 | 2 | 2048 | 2 | 256 | 320 | 448 | 640 |

## 3 EXPERIMENTAL RESULTS

In this section, we show that MOAT variants are effective on the ImageNet-1K (Russakovsky et al., 2015) image classification. We then deploy them to other recognition tasks, including COCO object detection (Lin et al., 2014), instance segmentation (Hariharan et al., 2014), and ADE20K (Zhou et al., 2019) semantic segmentation. MOAT can be seamlessly applied to downstream tasks. For small resolution inputs, we directly fine-tune the global attention, while for large resolution inputs, we simply convert the global attention to non-overlapping local window attention without using extra window-shifting mechanism. The detailed experiment setup could be found in the appendix.

Table 2: **Performance on ImageNet-1K**. **1K only:** Using ImageNet-1K only. **22K + 1K:** ImageNet-22K pretraining and ImageNet-1K fine-tuning. Tab. 8 shows comparisions with more SOTA methods and Tab. 9 reports the performances on ImageNet-1K-V2.

| | model | eval size | params | FLOPs | ImageNet-1K top-1 accuracy | |
| --- | --- | --- | --- | --- | --- | --- |
| | | | | | 1K only | 22K+1K |
| ConvNets | EfficientNetV2-L (Tan & Le, 2021) | $480^2$ | 120M | 53B | 85.7 | - |
| | EfficientNetV2-XL (Tan & Le, 2021) | $480^2$ | 208M | 94B | - | 87.3 |
| | ConvNeXt-T (Liu et al., 2022b) | $224^2$ | 29M | 4.5B | 82.1 | 82.9 |
| | ConvNeXt-L (Liu et al., 2022b) | $384^2$ | 198M | 101.0B | 85.5 | 87.5 |
| | ConvNeXt-XL (Liu et al., 2022b) | $384^2$ | 350M | 179.0B | - | 87.8 |
| ViTs | PVT-Large (Wang et al., 2021b) | $224^2$ | 61.4M | 9.8B | 81.7 | - |
| | Swin-T (Liu et al., 2021) | $224^2$ | 28M | 4.5B | 81.3 | - |
| | Swin-L (Liu et al., 2021) | $384^2$ | 197M | 103.9B | - | 87.3 |
| | SwinV2-L (Liu et al., 2021) | $384^2$ | 197M | 115.4B | - | 87.7 |
| | MViTv2-H (Li et al., 2022) | $512^2$ | 667M | 763.5B | - | 88.8 |
| Hybrid | PVTv2-B5 (Wang et al., 2022) | $224^2$ | 82M | 11.8B | 83.8 | - |
| | MaxViT-XL (Tu et al., 2022) | $512^2$ | 475M | 535.2B | - | 88.7 |
| | CoAtNet-0 (Dai et al., 2021) | $224^2$ | 25M | 4.2B | 81.6 | - |
| | CoAtNet-3 (Dai et al., 2021) | $384^2$ | 168M | 107.4B | 85.8 | 87.6 |
| | CoAtNet-4 (Dai et al., 2021) | $512^2$ | 275M | 360.9B | - | 88.6 |
| **Hybrid (ours)** | MOAT-0 | $224^2$ | 27.8M | 5.7B | 83.3 | 83.6 |
| | MOAT-1 | $224^2$ | 41.6M | 9.1B | 84.2 | 84.9 |
| | MOAT-2 | $224^2$ | 73.4M | 17.2B | 84.7 | 86.0 |
| | MOAT-3 | $224^2$ | 190.0M | 44.9B | 85.3 | 86.8 |
| | MOAT-0 | $384^2$ | 27.8M | 18.2B | 84.6 | 85.7 |
| | MOAT-1 | $384^2$ | 41.6M | 29.6B | 85.9 | 87.0 |
| | MOAT-2 | $384^2$ | 73.4M | 54.3B | 86.2 | 87.5 |
| | MOAT-3 | $384^2$ | 190.0M | 141.2B | 86.5 | 88.2 |
| | MOAT-1 | $512^2$ | 41.6M | 58.7B | 86.2 | 87.2 |
| | MOAT-2 | $512^2$ | 73.4M | 104.6B | 86.5 | 87.7 |
| | MOAT-3 | $512^2$ | 190.0M | 271.0B | 86.7 | 88.4 |
| | MOAT-4 | $512^2$ | 483.2M | 648.5B | - | 89.1 |

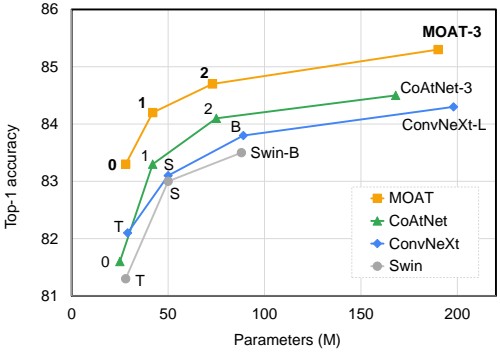

Figure 2: **Parameters vs. accuracy** using ImageNet-1K only with input size 224.

Figure 3: **FLOPs vs. accuracy** using ImageNet-1K only with input size 224.

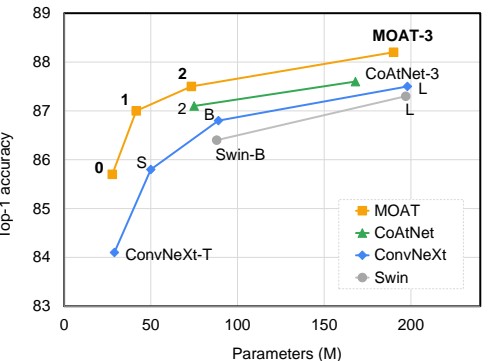

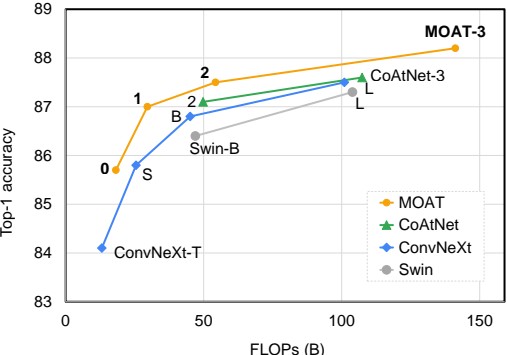

Figure 4: **Parameters vs. accuracy** using ImageNet-22K and ImageNet-1K with input size 384.

Figure 5: **FLOPs vs. accuracy** using ImageNet-22K and ImageNet-1K with input size 384.

**ImageNet Image Classification.** In Tab. 2, we include the current state-of-art methods in the categories of ConvNets, ViTs and Hybrid models. At similar model costs (parameters or FLOPs), our MOAT models consistently outperform all of them. Specifically, with the ImageNet-1K data only and input size 224, for light-weight models, our MOAT-0 significantly outperforms ConvNeXt-T (Liu et al., 2022b), Swin-T (Liu et al., 2022b), and CoAtNet-0 (Dai et al., 2021) by 1.2%, 2.0%, and 1.7%, respectively. For large-scale models using input size 384, MOAT-3 is able to surpass ConvNeXt-L, CoAtNet-3 by 1.0% and 0.7%, respectively. With the ImageNet-22K pretraining and input size 384, the prior arts ConvNeXt-L, Swin-L, and CoAtNet-3 already show strong performances (87.5%, 87.3% and 87.6%), while our MOAT-3 achieves the score of 88.2%, outperforming them by 0.7%, 0.9%, and 0.6%, respectively. For ImageNet-1K and input size 224, we plot the performances *vs.* parameters and FLOPs in Fig. 2 and Fig. 3, respectively. For ImageNet-22K pretraining and input size 384, we plot the performances *vs.* parameters and FLOPs in Fig. 4 and Fig. 5, respectively. In the figures, MOAT clearly demonstrates the best performance in all computation regimes. Finally, our largest model MOAT-4, with ImageNet-22K and input size 512, further attains 89.1% accuracy.

**COCO Detection.** Tab. 3 summarizes the COCO object detection (box) and instance segmentation (mask) results. Our MOAT backbones significantly outperform the baseline methods, including Swin (Liu et al., 2021) and ConvNeXt (Liu et al., 2022b) across different model sizes. Specifically, our MOAT-0 outperforms Swin-T and ConvNeXt-T by 5.4% and 5.5% $AP^{box}$ (3.7% and 3.7% $AP^{mask}$). Our MOAT-1 surpasses Swin-S and ConvNeXt-S by 5.9% and 5.8% $AP^{box}$ (4.3% and 4.0% $AP^{mask}$). Our MOAT-2, with 110M parameters, is still 5.5% and 4.5% $AP^{box}$ (3.5% and 2.4% $AP^{mask}$) better than Swin-B and ConvNeXt-B. Finally, our MOAT-3, using 227M parameters, achieves 59.2% $AP^{box}$ (50.3% $AP^{mask}$), setting a new state-of-the-art in the regime of model size 200M that is built on top of Cascade Mask R-CNN (Cai & Vasconcelos, 2018; He et al., 2017). More comparisons with smaller input size can be found in Tab. 12. **For tiny-MOAT**, tiny-MOAT-0/1 achieve the same performance as Swin-T/S and ConvNeXt-T/S but only use less than half of the parameters. Furthermore, tiny-MOAT-3 is pretrained with ImageNet-1K and attains 55.2 $AP^{box}$ with 57M parameters, surpassing the ImageNet-22k pretrained Swin-L (53.9 $AP^{box}$ with 254M parameters) and ConvNeXt-L (54.8 $AP^{box}$ with 255M parameters).

Table 3: **Object detection and instance segmentation on the COCO 2017 `val` set**. We employ Cascade Mask-RCNN, and single-scale inference (hard NMS). †: use ImageNet-22K pretrained weights. When using tiny-MOAT series as backbones, most of the model parameters come from the decoder. More comparisons at input size 896 is reported in Tab. 12.

| backbone | input size | params | FLOPs | $AP^{box}$ | $AP^{box}_{50}$ | $AP^{box}_{75}$ | $AP^{mask}$ | $AP^{mask}_{50}$ | $AP^{mask}_{75}$ |
|---|---|---|---|---|---|---|---|---|---|
| Swin-T | $1280 \times 800$ | 86M | 745B | 50.5 | 69.3 | 54.9 | 43.7 | 66.6 | 47.1 |
| Swin-S | $1280 \times 800$ | 107M | 838B | 51.8 | 70.4 | 56.3 | 44.7 | 67.9 | 48.5 |
| Swin-B† | $1280 \times 800$ | 145M | 982B | 53.0 | 71.8 | 57.5 | 45.8 | 69.4 | 49.7 |
| Swin-L† | $1280 \times 800$ | 254M | 1382B | 53.9 | 72.4 | 58.8 | 46.7 | 70.1 | 50.8 |
| ConvNeXt-T | $1280 \times 800$ | 86M | 741B | 50.4 | 69.1 | 54.8 | 43.7 | 66.5 | 47.3 |
| ConvNeXt-S | $1280 \times 800$ | 108M | 827B | 51.9 | 70.8 | 56.5 | 45.0 | 68.4 | 49.1 |
| ConvNeXt-B† | $1280 \times 800$ | 146M | 964B | 54.0 | 73.1 | 58.8 | 46.9 | 70.6 | 51.3 |
| ConvNeXt-L† | $1280 \times 800$ | 255M | 1354B | 54.8 | 73.8 | 59.8 | 47.6 | 71.3 | 51.7 |
| tiny-MOAT-0 | $1344 \times 1344$ | 41M | 612B | 50.5 | 69.3 | 56.0 | 43.3 | 66.6 | 47.3 |
| tiny-MOAT-1 | $1344 \times 1344$ | 42M | 628B | 51.9 | 71.6 | 56.1 | 44.6 | 68.4 | 48.3 |
| tiny-MOAT-2 | $1344 \times 1344$ | 47M | 669B | 53.0 | 72.2 | 58.0 | 45.0 | 69.4 | 48.8 |
| tiny-MOAT-3 | $1344 \times 1344$ | 57M | 754B | 55.2 | 74.8 | 60.6 | 47.0 | 71.8 | 51.2 |
| MOAT-0 | $1344 \times 1344$ | 65M | 799B | 55.9 | 73.9 | 60.9 | 47.4 | 70.9 | 52.1 |
| MOAT-1 | $1344 \times 1344$ | 79M | 921B | 57.7 | 76.0 | 63.4 | 49.0 | 73.4 | 53.2 |
| MOAT-2† | $1344 \times 1344$ | 110M | 1217B | 58.5 | 76.6 | 64.3 | 49.3 | 73.9 | 53.9 |
| MOAT-3† | $1344 \times 1344$ | 227M | 2216B | 59.2 | 77.8 | 64.9 | 50.3 | 74.8 | 55.5 |

**ADE20K Semantic Segmentation.** In Tab. 4, when using input size $513^2$, MOAT consistently outperforms the ConvNeXt counterparts. MOAT-0 surpasses ConvNeXt-T by 3.0% mIoU. Moreover, MOAT-2, with ImageNet-22k pretraining, surpasses ConvNeXt-B by 3.1%. The larger MOAT-3 and MOAT-4 further outperform ConvNeXt-L and ConvNeXt-XL by 4.9% and 5.4%, respectively. Finally, when using input size $641^2$, our MOAT-4 achieves the performance of 57.6% mIoU, setting a new state-of-the-art in the regime of models using input size $641^2$. **For tiny-MOAT**, tiny-MOAT-3 achieves comparable performance with ConvNeXt-S with less than half of the parameters.

**tiny-MOAT on ImageNet.** We simply scale down the channels of MOAT-0 to obtain the tiny-MOAT family without any specific adaptions. In the left of Tab. 5, with the similar model parameters,

Table 4: **Semantic segmentation on ADE20K `val` set.** We employ DeepLabv3+ (**single-scale** inference). Results for ConvNeXt and MOAT are obtained using the official code-base (Weber et al., 2021) with the same training recipe. †: use ImageNet-22K pretrained weights.

| backbone | input | params | FLOPs | mIoU (%) |
|---|---|---|---|---|
| ConvNeXt-T | $513^2$ | 34.2M | 47.6B | 45.8 |
| ConvNeXt-S | $513^2$ | 55.8M | 70.8B | 47.8 |
| ConvNeXt-B† | $513^2$ | 95.8M | 119.5B | 50.5 |
| ConvNeXt-L† | $513^2$ | 208.3M | 256.4B | 51.0 |
| ConvNeXt-XL† | $513^2$ | 364.0M | 446.2B | 51.8 |

| backbone | input | params | FLOPs | mIoU (%) |
|---|---|---|---|---|
| tiny-MOAT-0 | $513^2$ | 5.6M | 11.8B | 41.2 |
| tiny-MOAT-1 | $513^2$ | 7.8M | 15.2B | 43.1 |
| tiny-MOAT-2 | $513^2$ | 13.2M | 23.8B | 44.9 |
| tiny-MOAT-3 | $513^2$ | 24.2M | 41.2B | 47.5 |
| MOAT-0 | $513^2$ | 33.3M | 61.3B | 48.8 |
| MOAT-1 | $513^2$ | 47.0M | 85.4B | 51.8 |
| MOAT-2† | $513^2$ | 80.5M | 144.3B | 53.6 |
| MOAT-3† | $513^2$ | 198.4M | 331.5B | 55.9 |
| MOAT-4† | $513^2$ | 496.3M | 779.9B | 57.2 |
| MOAT-2† | $641^2$ | 80.5M | 242.0B | 54.7 |
| MOAT-3† | $641^2$ | 198.4M | 554.7B | 56.5 |
| MOAT-4† | $641^2$ | 496.3M | 1273.5B | 57.6 |

tiny-MOAT-0/1/2 surpass the Mobile-Former counterparts by 6.8%, 5.5%, and 4.3%, respectively. In the right of Tab. 5, our tiny-MOAT also shows stronger performances than MobileViT (Mehta & Rastegari, 2022a). Even compared with the concurrent work MobileViTv2 (Mehta & Rastegari, 2022b), tiny-MOAT-1/2/3 surpass their counterparts by 1.1%, 1.3%, and 2.1%, respectively.

Table 5: **Performances of tiny-MOAT family on ImageNet-1K.**

| input size $224^2$ | params | FLOPs | top-1 acc. |
|---|---|---|---|
| Mobile-Former-52M | 3.5M | 0.05B | 68.7 |
| Mobile-Former-96M | 4.6M | 0.1B | 72.8 |
| Mobile-Former-214M | 9.4M | 0.2B | 76.7 |
| Mobile-Former-508M | 14.0M | 0.5B | 79.3 |
| tiny-MOAT-0 | 3.4M | 0.8B | 75.5 |
| tiny-MOAT-1 | 5.1M | 1.2B | 78.3 |
| tiny-MOAT-2 | 9.8M | 2.3B | 81.0 |
| tiny-MOAT-3 | 19.5M | 4.5B | 82.7 |

| input size $256^2$ | params | FLOPs | top-1 acc. |
|---|---|---|---|
| MobileViT-XS | 2.3M | 0.7B | 74.8 |
| MobileViT-S | 5.6M | 2.0B | 78.4 |
| MobileViTv2-1.0 | 4.9M | 1.8B | 78.1 |
| MobileViTv2-1.5 | 10.6M | 4.0B | 80.4 |
| MobileViTv2-2.0 | 18.5M | 7.5B | 81.2 |
| tiny-MOAT-1 | 5.1M | 1.6B | 79.2 |
| tiny-MOAT-2 | 9.8M | 3.0B | 81.7 |
| tiny-MOAT-3 | 19.5M | 6.0B | 83.3 |

## 4  ABLATION STUDIES ON IMAGENET

At micro level, we perform ablation studies on the MOAT block design and downsampling layer in the following and the order of MBConv and Attention in MOAT block in section A.6.1. At macro level, we perform ablation studies on the MOAT-based model and MOAT meta architecture in section A.6.2.

**MOAT block design.** In Tab. 6, we ablate the MOAT block design, which only affects the last two stages of MOAT, and we keep everything else the same (*e.g.*, training recipes). We start from the Transformer block, consisting of Attn (self-attention) and MLP, which already attains a strong top-1 accuracy (82.6%). Directly inserting a $3 \times 3$ depthwise convolution in the MLP degrades the performance by 0.9%. If we additionally insert batch normalization and GeLU between convolutions (*i.e.*, replace MLP with MBConv, but no Squeeze-and-Excitation), the performance is improved to 82.9%. Finally, placing MBConv before Attn reaches the performance of 83.3%. Additionally, our MOAT block brings more improvements (from 1.2% up to 2.6% gains) in the tiny model regime.

**Downsampling layer.** For the MOAT block design, we do not need the extra downsampling layer like (1) average-pooling in CoAtNet (Dai et al., 2021), (2) patch-embedding layer (*i.e.*, $2 \times 2$ convolution with stride 2) in Swin (Liu et al., 2021) and ConvNeXt (Liu et al., 2022b), or (3) strided depthwise convolution in PiT (Heo et al., 2021) and RegionViT (Chen et al., 2022a). As shown in Tab. 7, using patch-embedding layer indeed improves over the average-pooling scheme by 0.2% accuracy, but it takes more cost of model parameters. Additionally, using the strided depthwise convolution for downsampling leads to 0.2% worse performance than the patch-embedding layer. By contrast, our MOAT design (*i.e.*, delegating the downsampling to the MBConv block) shows the best performance with the least cost of parameters and comparable FLOPs.

Table 6: **Ablation studies of MOAT block design** on ImageNet-1K with input size 224.

| model | block composition | params | FLOPs | top-1 acc. |
|---|---|---|---|---|
| MOAT-0 | Attn + MLP | 28.0M | 5.4B | 82.6 |
| | Attn + MLP (w/ depth. conv) | 28.2M | 5.4B | 81.7 |
| | Attn + MBConv | 28.2M | 5.4B | 82.9 |
| | MBConv + Attn | 27.8M | 5.7B | 83.3 |

| model | block composition | params | FLOPs | top-1 acc. |
|---|---|---|---|---|
| tiny-MOAT-2 | Attn + MLP | 9.8M | 2.2B | 79.8 |
| | MBConv + Attn | 9.8M | 2.3B | 81.0 |
| tiny-MOAT-1 | Attn + MLP | 5.1M | 1.1B | 76.2 |
| | MBConv + Attn | 5.1M | 1.2B | 78.3 |
| tiny-MOAT-0 | Attn + MLP | 3.3M | 0.8B | 72.9 |
| | MBConv + Attn | 3.4M | 0.8B | 75.5 |

Table 7: **Ablation studies of the downsampling layer design** on ImageNet-1K, using MOAT-0 and input size 224. We compare our MOAT design (in grey) with (1) CoAtNet (using average-pooling for downsampling), (2) Swin/ConvNeXt designs (using strided $2 \times 2$ convolution for downsampling), and (3) PiT/RegionViT designs (using strided $3 \times 3$ depthwise convolution for downsampling).

| block composition | downsampling type | params (M) | FLOPs (B) | top-1 acc. |
|---|---|---|---|---|
| AveragePooling + Attn + MLP | CoAtNet | 28.0 | 5.4 | 82.6 |
| PatchEmbedding + Attn + MLP | Swin, ConvNeXt | 30.2 | 5.6 | 82.8 |
| StridedDepthConv + Attn + MLP | PiT, RegionVit | 28.8 | 5.5 | 82.6 |
| MBConv + Attn | MOAT | 27.8 | 5.7 | 83.3 |

## 5 RELATED WORK

Transformers (Vaswani et al., 2017) were recently introduced to the vision community (Wang et al., 2018; Ramachandran et al., 2019; Hu et al., 2019) and demonstrated remarkable performance on vision recognition tasks (Carion et al., 2020; Zhu et al., 2021; Wang et al., 2021a; Arnab et al., 2021; Liu et al., 2021; Cheng et al., 2021; Yu et al., 2022a; Kim et al., 2022; Cheng et al., 2022; Yu et al., 2022b), thanks to their ability to efficiently encode long-range interaction via the attention mechanism (Bahdanau et al., 2015). Particularly, ViT (Dosovitskiy et al., 2021) obtains impressive results on ImageNet (Russakovsky et al., 2015) by applying the vanilla Transformer with the novel large stride patch embedding, after pretraining on the proprietary large-scale JFT dataset (Sun et al., 2017). There have been several works aiming to improve the vision transformers, either with better training strategies (Touvron et al., 2021a;b; Steiner et al., 2021; Zhai et al., 2022; Touvron et al., 2022) or with efficient local-attention modules (Huang et al., 2019; Ho et al., 2019; Wang et al., 2020; Liu et al., 2021; Chu et al., 2021; Yang et al., 2021; Yu et al., 2021; Dong et al., 2022; Tu et al., 2022).

Since the debut of AlexNet (Krizhevsky et al., 2012), the vision community has witnessed a rapid improvement on the ImageNet benchmark using different types of ConvNets, including (but not limited to) VGGNet (Simonyan & Zisserman, 2015), Inceptions (Szegedy et al., 2015; Ioffe & Szegedy, 2015; Szegedy et al., 2016; 2017), ResNets (He et al., 2016a;b), ResNeXt (Xie et al., 2017), DenseNet (Huang et al., 2017), SENet (Hu et al., 2018), MobileNets (Howard et al., 2017; Sandler et al., 2018; Howard et al., 2019), EfficientNets (Tan & Le, 2019; 2021), and ConvNeXt (Liu et al., 2022b) each focusing on different aspects of accuracy and efficiency. The ubiquity of ConvNets in computer vision could be attributed to their built-in inductive biases.

Given the success of Transformers and ConvNets, another line of research is to explore how to effectively combine them. Swin (Liu et al., 2021; 2022a), PVT (Wang et al., 2021b; 2022), MViT (Fan et al., 2021; Li et al., 2022), and PiT (Heo et al., 2021) adopt the ConvNet hierarchical structure to extract multi-scale features for Transformers. SASA (Ramachandran et al., 2019), AA-ResNet (Bello et al., 2019), Axial-ResNet (Wang et al., 2020) and BoTNet (Srinivas et al., 2021) incorporate the attention modules to ResNets. CvT (Wu et al., 2021), LeViT (Graham et al., 2021), Visformer (Chen et al., 2021b), and ViT$_C$ (Xiao et al., 2021) replace ViT's patch embedding with strided convolutions. CeiT (Yuan et al., 2021a) and CMT (Guo et al., 2022) incorporate depthwise convolution to the transformer block's MLP. ViTAE (Xu et al., 2021) adopts parallel attention modules and convolutional layers. LVT (Yang et al., 2022) introduces local self-attention into the convolution. Recently, CoAtNet (Dai et al., 2021) and MobileViT (Mehta & Rastegari, 2022a) propose hybrid models that build on top of the efficient Mobile Convolution (Sandler et al., 2018) and Transformer block.

**Acknowledgements** We thank Wen-Sheng Chu for the support and discussion. We gratefully acknowledge supports from the Office of Naval Research. N00014-21-1-2812.

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

# A   APPENDIX

In the appendix, we provide more details for both our model and experiments.

- In section A.1, we provide MOAT implementation details.
- In section A.2.1, we provide ImageNet experimental details.
- In section A.2.2, we provide ImageNet-V2 experimental results.
- In section A.3.1, we provide COCO detection experimental details.
- In section A.3.2, we provide more COCO object detection experimental results.
- In section A.4, we provide ADE20K semantic segmentation experimental detaills.
- In section A.5, we provide COCO panoptic segmentation experiments.
- In section A.6.1, we provide ablation studies on the MOAT micro-level design.
- In section A.6.2, we provide ablation studies on the MOAT macro-level design.
- In section A.7, we provide the ImageNet trainng time, peak training memory and throughput measurement of MOAT models.
- In section A.8, we discuss limitations of our model.

## A.1   MOAT IMPLEMENTATION DETAILS

In the MOTA networks, we employ kernel size 3 for both convolutions and depthwise convolutions. We use the multi-head self attention (Vaswani et al., 2017), where each attention head has channels 32. For the MBConv and MOAT blocks, we use expansion ratio 4. The SE module (Hu et al., 2018) in the MBConv blocks (*i.e.*, 2nd and 3rd stages) adopt reduction ratio 4 (relative to the input channels).

Our MOAT block includes the relative positional embedding (Shaw et al., 2018; Dai et al., 2021) for ImageNet. However, the downstream tasks usually take a larger input resolution than ImageNet, demanding for a special adaptation (*e.g.*, bilinear interpolation of pretrained positional embedding). For simplicity, we remove the positional embedding, when running MOAT on downstream tasks.

## A.2   IMAGENET IMAGE CLASSIFICATION

### A.2.1   IMAGENET EXPERIMENTS

The ImageNet-1K dataset (Russakovsky et al., 2015) contains 1.2M training images with 1000 classes. We report top-1 accuracy on the ImageNet-1K validation set, using the last checkpoint. We also experiment with pretraining on the larger ImageNet-22K dataset, and then fine-tuning on the ImageNet-1K. We closely follow the prior works (Dai et al., 2021; Liu et al., 2022b) and provide more details below. In Tab. 8, we compare our MOAT with more state-of-the-art models.

**Experimental setup.** We train MOAT models on ImageNet-1K with resolution 224 for 300 epochs. If pretraining on the larger ImageNet-22K, we use resolution 224 and 90 epochs. Afterwards, the models are fine-tuned on ImageNet-1K for 30 epochs. During fine-tuning, we also experiment with larger resolutions (*e.g.*, 384 and 512). We employ the typical regularization methods during training, such as label smoothing (Szegedy et al., 2016), RandAugment (Cubuk et al., 2020), MixUp (Zhang et al., 2017), stochastic depth (Huang et al., 2016), and Adam (Kingma & Ba, 2015) with decoupled weight decay (*i.e.*, AdamW (Loshchilov & Hutter, 2019)). See Tab. 10 and Tab. 11 for detailed hyper-parameters.

### A.2.2   IMAGENET-1K-V2 EVALUATION

To further demonstrate the transferability and generalizability of our MOAT models, we perform additional evaluations on the ImageNet-1K-V2 (Recht et al., 2019), using our ImageNet (Russakovsky et al., 2015) pretrained checkpoints. We report an extensive evaluation, using MOAT and several input resolutions, on ImageNet-1K-V2, aiming to establish another solid baseline for the community, as we notice that most of the existing models do not report results on ImageNet-1K-V2. As shown in the Tab. 9, MOAT does not overfit to ImageNet-1K-V1 dataset and generalizes well to ImageNet-1K-V2 dataset, as we observe a continuous performance improvement from small to large models.

Table 8: **Performance on ImageNet-1K** with more state-of-the-art models are included. **1K only:** Using ImageNet-1K only. **22K + 1K:** ImageNet-22K pretraining and ImageNet-1K fine-tuning.

| | model | eval size | params | FLOPs | ImageNet-1K top-1 accuracy | |
|---|---|---|---|---|---|---|
| | | | | | 1K only | 22K+1K |
| ConvNets | RegNetY-16G (Radosavovic et al., 2020) | $224^2$ | 84M | 16.0B | 82.9 | - |
| | NFNet-F5 (Brock et al., 2021) | $544^2$ | 377M | 289.8B | 86.0 | - |
| | EfficientNetV2-S (Tan & Le, 2021) | $480^2$ | 22M | 8.8B | 83.9 | 84.9 |
| | EfficientNetV2-M (Tan & Le, 2021) | $480^2$ | 54M | 24B | 85.1 | 86.2 |
| | EfficientNetV2-L (Tan & Le, 2021) | $480^2$ | 120M | 53B | 85.7 | - |
| | EfficientNetV2-XL (Tan & Le, 2021) | $480^2$ | 208M | 94B | - | 87.3 |
| | ConvNeXt-T (Liu et al., 2022b) | $224^2$ | 29M | 4.5B | 82.1 | 82.9 |
| | ConvNeXt-S (Liu et al., 2022b) | $224^2$ | 50M | 8.7B | 83.1 | 84.6 |
| | ConvNeXt-B (Liu et al., 2022b) | $224^2$ | 89M | 15.4B | 83.8 | 85.8 |
| | ConvNeXt-L (Liu et al., 2022b) | $384^2$ | 198M | 101.0B | 85.5 | 87.5 |
| | ConvNeXt-XL (Liu et al., 2022b) | $384^2$ | 350M | 179.0B | 85.5 | 87.8 |
| ViTs | DeiT-B (Touvron et al., 2021a) | $384^2$ | 86M | 55.4B | 83.1 | - |
| | CaiT-S-36 (Touvron et al., 2021b) | $384^2$ | 68M | 48.0B | 85.0 | - |
| | DeepViT-L (Zhou et al., 2021) | $224^2$ | 55M | 12.5B | 83.1 | - |
| | PVT-Large (Wang et al., 2021b) | $224^2$ | 61.4M | 9.8B | 81.7 | - |
| | HaloNet-H4 (Vaswani et al., 2021) | $384^2$ | 85M | - | 85.6 | - |
| | HaloNet-H5 (Vaswani et al., 2021) | $512^2$ | 85M | - | 85.8 | - |
| | Swin-T (Liu et al., 2021) | $224^2$ | 28M | 4.5B | 81.3 | - |
| | Swin-S (Liu et al., 2021) | $224^2$ | 50M | 8.7B | 83.0 | - |
| | Swin-B (Liu et al., 2021) | $224^2$ | 88M | 15.4B | 83.5 | 85.2 |
| | Swin-L (Liu et al., 2021) | $384^2$ | 197M | 103.9B | - | 87.3 |
| | SwinV2-L (Liu et al., 2022a) | $384^2$ | 197M | 115.4B | - | 87.7 |
| | Focal-B (Yang et al., 2021) | $224^2$ | 89.8M | 16.0B | 83.8 | - |
| | CSwin-B (Dong et al., 2022) | $384^2$ | 78M | 47.0B | 85.4 | 87.0 |
| | CSwin-L (Dong et al., 2022) | $384^2$ | 173M | 96.8B | - | 87.5 |
| | MViTv2-H (Li et al., 2022) | $512^2$ | 667M | 763.5B | - | 88.8 |
| Hybrid | BotNet-T7 (Srinivas et al., 2021) | $384^2$ | 75.1M | 45.8B | 84.7 | - |
| | LambdaResNet-420 (Bello, 2021) | $320^2$ | - | - | 84.9 | - |
| | T2T-ViT-24 (Yuan et al., 2021b) | $224^2$ | 64.1M | 15.0B | 82.6 | - |
| | CMT-S (Guo et al., 2022) | $224^2$ | 25.1M | 4.0B | 83.5 | - |
| | CeiT-S (Yuan et al., 2021a) | $384^2$ | 24.2M | 12.9B | 83.3 | - |
| | CvT-21 (Wu et al., 2021) | $384^2$ | 32M | 24.9B | 83.3 | - |
| | PVTv2-B5 (Wang et al., 2022) | $224^2$ | 82M | 11.8B | 83.8 | - |
| | MaxViT-XL (Tu et al., 2022) | $512^2$ | 475M | 535.2B | - | 88.7 |
| | CoAtNet-0 (Dai et al., 2021) | $224^2$ | 25M | 4.2B | 81.6 | - |
| | CoAtNet-1 (Dai et al., 2021) | $224^2$ | 42M | 8.4B | 83.3 | - |
| | CoAtNet-2 (Dai et al., 2021) | $224^2$ | 75M | 15.7B | 84.1 | - |
| | CoAtNet-3 (Dai et al., 2021) | $384^2$ | 168M | 107.4B | 85.8 | 87.6 |
| | CoAtNet-4 (Dai et al., 2021) | $512^2$ | 275M | 360.9B | - | 88.6 |
| **Hybrid (ours)** | MOAT-0 | $224^2$ | 27.8M | 5.7B | 83.3 | 83.6 |
| | MOAT-1 | $224^2$ | 41.6M | 9.1B | 84.2 | 84.9 |
| | MOAT-2 | $224^2$ | 73.4M | 17.2B | 84.7 | 86.0 |
| | MOAT-3 | $224^2$ | 190.0M | 44.9B | 85.3 | 86.8 |
| | MOAT-0 | $384^2$ | 27.8M | 18.2B | 84.6 | 85.7 |
| | MOAT-1 | $384^2$ | 41.6M | 29.6B | 85.9 | 87.0 |
| | MOAT-2 | $384^2$ | 73.4M | 54.3B | 86.2 | 87.5 |
| | MOAT-3 | $384^2$ | 190.0M | 141.2B | 86.5 | 88.2 |
| | MOAT-1 | $512^2$ | 41.6M | 58.7B | 86.2 | 87.2 |
| | MOAT-2 | $512^2$ | 73.4M | 104.6B | 86.5 | 87.7 |
| | MOAT-3 | $512^2$ | 190.0M | 271.0B | 86.7 | 88.4 |
| | MOAT-4 | $512^2$ | 483.2M | 648.5B | - | 89.1 |

Under the fair comparison, with ImageNet-22K pretrainng and input size 384, MOAT-2/3 surpass the current state-of-the-art model SwinV2-B/L by 0.6/1.7%, respectively. Additionally, our MOAT-4, with input size 512, achieves a new state-of-the-art performance of 81.5%, without extra proprietary training data.

Table 9: **Performance on ImageNet-1K-V2**.

| model | params | input size | FLOPs | ImageNet-1K-V2 top-1 accuracy (%) | |
|---|---|---|---|---|---|
| | | | | 1K only | 22K + 1K |
| LeViT-256 (Graham et al., 2021) | 256 | 18.9M | 1.1B | 70.0 | – |
| DeiT-B (Touvron et al., 2021a) | 224 | 86M | 17.5B | 71.5 | – |
| CaiT-S36 (Touvron et al., 2021b) | 224 | 68M | 13.9B | 72.5 | – |
| SwinV2-B (Liu et al., 2021) | 384 | 88M | 54.7B | – | 78.1 |
| SwinV2-L (Liu et al., 2021) | 384 | 197M | 115.4B | – | 78.3 |
| tiny-MOAT-0 | 224 | 3.4M | 0.8B | 64.3 | – |
| tiny-MOAT-1 | 224 | 5.1M | 1.2B | 67.3 | – |
| tiny-MOAT-2 | 224 | 9.8M | 2.3B | 70.1 | – |
| tiny-MOAT-3 | 224 | 19.5M | 4.5B | 72.1 | – |
| tiny-MOAT-1 | 256 | 5.1M | 1.6B | 68.2 | – |
| tiny-MOAT-2 | 256 | 9.8M | 3.0B | 70.9 | – |
| tiny-MOAT-3 | 256 | 19.5M | 6.0B | 72.9 | – |
| MOAT-0 | 224 | 27.8M | 5.7B | 72.8 | 74.1 |
| MOAT-1 | 224 | 41.6M | 9.1B | 74.2 | 75.8 |
| MOAT-2 | 224 | 73.4M | 17.2B | 74.3 | 76.7 |
| MOAT-3 | 224 | 190.0M | 44.9B | 75.5 | 78.4 |
| MOAT-0 | 384 | 27.8M | 18.2B | 74.5 | 76.4 |
| MOAT-1 | 384 | 41.6M | 29.6B | 76.2 | 78.1 |
| MOAT-2 | 384 | 73.4M | 54.3B | 76.5 | 78.7 |
| MOAT-3 | 384 | 190.0M | 141.2B | 77.5 | 80.0 |
| MOAT-1 | 512 | 41.6M | 58.7B | 76.8 | 78.4 |
| MOAT-2 | 512 | 73.4M | 104.6B | 77.1 | 79.3 |
| MOAT-3 | 512 | 190.0M | 271.0B | 77.8 | 80.6 |
| MOAT-4 | 512 | 483.2M | 648.5B | – | 81.5 |

Table 10: **MOAT ImageNet hyper-parameter settings.**

| hyper-parameter | ImageNet-1K | | ImageNet-22K | |
|---|---|---|---|---|
| | 1K pre-training (MOAT-0/1/2/3) | 1K → 1K fine-tuning | 22K pre-training (MOAT-0/1/2/3) | 22K → 1K fine-tuning |
| stochastic depth rate | 0.2 / 0.3 / 0.5 / 0.7 | 0.2 / 0.3 / 0.5 / 0.9 | 0.1 / 0.2 / 0.3 / 0.6 | 0.1 / 0.2 / 0.3 / 0.6 |
| center crop | true | false | true | false |
| randaugment | 2, 15 | 2, 15/15/15/20 | 2, 5 | 2, 5 |
| mixup alpha | 0.8 | 0.8 | none | none |
| loss type | softmax | softmax | sigmoid | softmax |
| label smoothing | 0.1 | 0.1 | 0.0001 | 0.1 |
| train epochs | 300 | 30 | 90 | 30 |
| train batch size | 4096 | 512 | 4096 | 1024 |
| optimizer type | AdamW | AdamW | AdamW | AdamW |
| peak learning rate | 3e-3 | 5e-5 | 1e-3 | 5e-5 |
| min learning rate | 1e-5 | 5e-5 | 1e-5 | 5e-5 |
| warm-up | 10K steps | none | 5 epochs | none |
| lr decay schedule | cosine | none | linear | none |
| weight decay rate | 0.05 | 1e-8 | 0.01 | 1e-8 |
| gradient clip | 1.0 | 1.0 | 1.0 | 1.0 |
| EMA decay rate | 0.9999 | 0.9999 | None | 0.9999 |

## A.3  COCO OBJECT DETECTION AND INSTANCE SEGMENTATION

### A.3.1  COCO OBJECT DETECTION EXPERIMENTAL DETAILS

**Experimental setup.** We train Cascade Mask R-CNN (Cai & Vasconcelos, 2018; He et al., 2017) on the COCO 2017 dataset (Lin et al., 2014) with our MOAT architectures. The dataset contains 118K training and 5K validation samples. We use the official TensorFlow (Abadi et al., 2016) implementation of Cascade Mask R-CNN by TF-Vision Model Garden (Yu et al., 2020). Our training setting closely follows the prior works (Chen et al., 2022b; Tu et al., 2022), except that we use batch size 64 and initial learning rate 0.0001. To adapt the MOAT models to high-resolution inputs, we

Table 11: **tiny-MOAT ImageNet hyper-parameter settings.** $^\star$: use EMA decay rate 0.9999 for tiny-MOAT-3.

| hyper-parameter | ImageNet-1K | |
| --- | --- | --- |
| | 1K input size 224 (tiny-MOAT-0/1/2/3) | 1K input size 256 |
| stochastic depth rate | 0.0 / 0.0 / 0.0 / 0.1 | 0.0 / 0.0 / 0.0 / 0.1 |
| center crop | true | true |
| randaugment | 2, 15 | 2, 15 |
| mixup alpha | 0.8 | 0.8 |
| loss type | softmax | softmax |
| label smoothing | 0.1 | 0.1 |
| train epochs | 300 | 300 |
| train batch size | 4096 | 4096 |
| optimizer type | AdamW | AdamW |
| peak learning rate | 3e-3 | 3e-3 |
| min learning rate | 1e-5 | 1e-5 |
| warm-up | 10K steps | 10K steps |
| lr decay schedule | cosine | cosine |
| weight decay rate | 0.05 | 0.05 |
| gradient clip | 1.0 | 1.0 |
| EMA decay rate | None$^\star$ | None$^\star$ |

partition the features into non-overlapping windows for the self-attention computations with the window size set to 14 for the second last stage, and use global attention for the last stage. As a result of this window partition, the input size must be divisible by 14. The TF-Vision Model Garden codebase further requires the input size to be square (with padding) and divisible by 64. Hence, we choose 1344 as the input size, similar to the size used in the baseline methods (*i.e.*, longest side is no more than 1333). We use Feature Pyramid Network (Lin et al., 2017) to integrate features from different levels.

### A.3.2 MORE COCO OBJECT DETECTION EXPERIMENTAL RESULTS

In this section, we perform more COCO object detection experiments with 896 input size. All the backbone are pretrained on ImageNet-1K dataset. MOAT-0/1/2 surpass UViT (Chen et al., 2021a) and MaxViT (Tu et al., 2022) by 3.9/5.2/4.9% $AP^{box}$ (3.1/4.1/3.9% $AP^{mask}$), and 3.0/4.0/4.0% $AP^{box}$ (2.4/3.2/3.0% $AP^{mask}$), respectively.

Table 12: **Object detection and instance segmentation on the COCO 2017 `val` set**. We employ Cascade Mask-RCNN, and single-scale inference (hard NMS). All backbones are pretrained on ImageNet-1K.

| backbone | input size | params | FLOPs | $AP^{box}$ | $AP^{box}_{50}$ | $AP^{box}_{75}$ | $AP^{mask}$ | $AP^{mask}_{50}$ | $AP^{mask}_{75}$ |
| --- | --- | --- | --- | --- | --- | --- | --- | --- | --- |
| UViT-T (Chen et al., 2021a) | $896 \times 896$ | 51M | 720B | 51.2 | – | – | 43.9 | – | – |
| UViT-S (Chen et al., 2021a) | $896 \times 896$ | 59M | 882B | 51.9 | – | – | 44.5 | – | – |
| UViT-B (Chen et al., 2021a) | $896 \times 896$ | 74M | 1160B | 52.5 | – | – | 44.8 | – | – |
| MaxViT-T (Tu et al., 2022) | $896 \times 896$ | 86M | 475B | 52.1 | 71.9 | 56.8 | 44.6 | 69.1 | 48.4 |
| MaxViT-S (Tu et al., 2022) | $896 \times 896$ | 108M | 595B | 53.1 | 72.5 | 58.1 | 45.4 | 69.8 | 49.5 |
| MaxViT-B (Tu et al., 2022) | $896 \times 896$ | 146M | 856B | 53.4 | 72.9 | 58.1 | 45.7 | 70.3 | 50.0 |
| MOAT-0 | $896 \times 896$ | 65M | 525B | 55.1 | 73.6 | 59.9 | 47.0 | 70.5 | 51.1 |
| MOAT-1 | $896 \times 896$ | 79M | 580B | 57.1 | 75.7 | 62.6 | 48.6 | 72.9 | 52.7 |
| MOAT-2 | $896 \times 896$ | 110M | 710B | 57.4 | 76.0 | 63.0 | 48.7 | 73.2 | 53.1 |

### A.4 ADE20K SEMANTIC SEGMENTATION

**Experimental setup.** We experiment with the proposed MOAT models on ADE20K semantic segmentation dataset (Zhou et al., 2019) using DeepLabv3+ (Chen et al., 2018; 2017). We fine-tune the global attention for MOAT. The same training strategies are used for all backbone variants. Specifically, for training hyper-parameters, we train the model with 32 TPU cores for 180k iterations, with batch size 64, Adam (Kingma & Ba, 2015) optimizer, and a poly schedule learning rate starting at 0.0001. For data augmentations, the inputs images are resized and padded to either $513 \times 513$ or

$641 \times 641$, with random cropping, flipping, and color jittering (Cubuk et al., 2019). **No test-time augmentation is used during inference.**

## A.5 COCO PANOPTIC SEGMENTATION

**Experimental setup.** We also evaluate the proposed MOAT architectures on the challenging COCO panoptic segmentation dataset (Lin et al., 2014) using Panoptic-DeepLab (Cheng et al., 2020) with the official codebase (Weber et al., 2021). We fine-tune the global attention on downstream segmentation tasks for MOAT. We adopt the same training strategies for MOAT and its counterparts. Specifically, for training hyper-parameters, we train the model with 32 TPU cores for 200k iterations with the first 2k for warm-up stage. We use batch size 64, Adam (Kingma & Ba, 2015) optimizer, and a poly schedule learning rate starting at 0.0005. For data augmentations, the inputs images are resized and padded to $641 \times 641$, with random cropping, flipping, and color jittering (Cubuk et al., 2019). **No test-time augmentation is used during inference.**

**Main results.** The results are summarized in Tab. 13, where MOAT consistently outperforms other backbones. Specifically, our MOAT-0 surpasses ConvNeXt-T significantly by 4.3% PQ. In the large model regime, MOAT-3 surpasses ConvNeXt-L by 3.5%. Our MOAT-4 achieves the performance of 46.7% PQ, outperforming the heavy backbone SWideRNet (Chen et al., 2020) by 2.3%.

Table 13: **Panoptic segmentation on COCO `val` set.** The results are obtained by applying different backbones with Panoptic-DeepLab, using **single-scale** inference (*i.e.*, no test-time augmentation). Results for MobileNet, ResNet, and Xception are cited from (Cheng et al., 2020), and results for SWideRNet is cited from (Chen et al., 2020), while results for ConvNeXt and MOAT are obtained using the official code-base (Weber et al., 2021) with the same training recipe. All models are trained and evaluated with input images resized to $641 \times 641$, and thus FLOPs are also measured w.r.t. size $641 \times 641$. †: use ImageNet-22K pretrained weights.

| backbone | params | FLOPs | PQ (%) | PQ$^{Th}$ (%) | PQ$^{St}$ (%) |
|---|---|---|---|---|---|
| MobileNet-V3 (Howard et al., 2019) | - | 12.2B | 30.0 | - | - |
| ResNet50 (He et al., 2016a) | - | 77.8B | 35.1 | - | - |
| Xception-71 (Chollet, 2017) | - | 109.2B | 38.9 | - | - |
| ConvNeXt-T (Liu et al., 2022b) | 40.3M | 51.3B | 36.7 | 37.3 | 35.7 |
| ConvNeXt-S (Liu et al., 2022b) | 61.9M | 87.2B | 40.0 | 41.4 | 37.9 |
| ConvNeXt-B† (Liu et al., 2022b) | 103.8M | 146.2B | 41.7 | 43.6 | 38.9 |
| ConvNeXt-L† (Liu et al., 2022b) | 220.1M | 312.8B | 41.9 | 43.6 | 39.4 |
| ConvNeXt-XL† (Liu et al., 2022b) | 379.6M | 544.1B | 43.0 | 44.9 | 40.0 |
| SWideRNet (Chen et al., 2020) | 752.5M | 2614.0B | 44.4 | - | - |
| MOAT-0 | 39.5M | 76.8B | 41.0 | 42.6 | 38.6 |
| MOAT-1 | 53.1M | 119.7B | 43.0 | 44.7 | 40.4 |
| MOAT-2† | 88.5M | 199.7B | 43.9 | 45.9 | 40.8 |
| MOAT-3† | 208.3M | 493.3B | 45.4 | 48.3 | 41.1 |
| MOAT-4† | 512.0M | 1134.7B | 46.7 | 49.5 | 42.4 |

## A.6 MORE ABLATION STUDIES

### A.6.1 ABLATION STUDIES ON THE MOAT MICRO-LEVEL DESIGN

**Order of MBConv and Attn in MOAT block.** Our MOAT block design reverses the order of Attention (Attn) and Mobile Convolution (MBConv), delegating the downsampling duty to the strided depthwise convolution within the MBConv. However, the dowsampling can be still performed in the MBConv with the *original* order (*i.e.*, Attn + MBConv). Since the operations, Attn and MBConv, are interlaced, the key difference then comes from the first block in each stage, where the Attn is operated on the (1) spatially downsampled and/or (2) channel expanded features. To conduct the study, we employ different blocks in the MOAT variants, using "Attn + MLP", "Attn + MBConv", or "MBConv + Attn". For the "Attn + MBConv" block, we further ablate the place (Attn *vs*. MBConv), where we apply the spatial downsampling and channel expansion operations.

In Tab. 14, we observe the following results. First, replacing the MLP with MBConv improves the performance by 0.3% and 0.7% for MOAT-0 and tiny-MOAT-2. Second, if we perform both spatial downsampling and channel expansion at the MBConv block, the performance is further improved by 0.5% and 0.9% for MOAT-0 and tiny-MOAT-2, showing that MBConv learns better downsampled

Table 14: **Ablation studies of the order of MBConv and Attention (Attn)** on ImageNet-1K with input 224. We also ablate the place, where we apply the spatial downsampling and channel expansion.

| model | block composition | spatial downsampling | channel expansion | params (M) | FLOPs (B) | top-1 acc. |
|---|---|---|---|---|---|---|
| | Attn + MLP | Attn | Attn | 28.0 | 5.4 | 82.6 |
| | Attn + MBConv | Attn | Attn | 28.2 | 5.4 | 82.9 |
| MOAT-0 | Attn + MBConv | MBConv | MBConv | 25.6 | 5.8 | 83.1 |
| | MBConv + Attn | MBConv | MBConv | 27.8 | 5.7 | 83.3 |
| | Attn + MBConv | MBConv | Attn | 29.3 | 7.1 | 83.2 |
| | Attn + MLP | Attn | Attn | 9.8 | 2.2 | 79.8 |
| | Attn + MBConv | Attn | Attn | 9.9 | 2.2 | 80.5 |
| tiny-MOAT-2 | Attn + MBConv | MBConv | MBConv | 9.0 | 2.3 | 80.7 |
| | MBConv + Attn | MBConv | MBConv | 9.8 | 2.3 | 81.0 |
| | Attn + MBConv | MBConv | Attn | 10.3 | 2.8 | 81.0 |

features. However, this design is equivalent to shifting the first Attn layer to its previous stage, reducing the representation capacity of the current stage. More concretely, only the last stage will be affected, since one layer is shifted. Third, to enhance the representation capacity, reversing the order of Attn and MBConv allows us to keep the first Attn layer in the same stage. This design further improves the performance by 0.7% and 1.2% for MOAT-0 and tiny-MOAT-2. Fourth, to compensate for the shifting effect, we could also employ another $1 \times 1$ convolution to expand the channels at the first Attn layer (then, MBConv only performs the spatial downsampling). However, this design performs similarly to our MOAT block design, but uses more parameters and FLOPs.

### A.6.2 ABLATION STUDIES ON THE MOAT MACRO-LEVEL DESIGN

**Ablation studies on MOAT-based model.** In Tab. 15, we ablate the stage-wise design by using either MBConv or MOAT block in stage 2 to stage 5. The first stage is the convolutional stem, containing two $3 \times 3$ convolutions. We use the layer layout of MOAT-0. As shown in the table, the pure MOAT-based model (*i.e.*, using MOAT blocks for all four stages) achieves the best performance of 83.6%, which however uses the most FLOPs. Our MOAT model design (*i.e.*, use MOAT block in the last two stages) attains the better trade-off between accuracy and model complexity.

Table 15: **Ablation studies of MOAT-based model** on ImageNet-1K, using MOAT-0 layer layout and input size 224. We change the block type (MBConv *vs.* MOAT block) from stage 2 to stage 5. The first stage is fixed to use the convolutional stem.

| stage-2 | stage-3 | stage-4 | stage-5 | params (M) | FLOPs (B) | top-1 acc. |
|---|---|---|---|---|---|---|
| MOAT | MOAT | MOAT | MOAT | 28.2 | 11.9 | 83.6 |
| MBConv | MOAT | MOAT | MOAT | 28.1 | 6.9 | 83.5 |
| MBConv | MBConv | MOAT | MOAT | 27.8 | 5.7 | 83.3 |
| MBConv | MBConv | MBConv | MOAT | 25.7 | 4.7 | 82.2 |
| MBConv | MBConv | MBConv | MBConv | 23.4 | 4.5 | 82.0 |

**Ablation studies on MOAT meta architecture.** We perform ablation studies on the meta-architecture by varying the number of blocks per stage. For simplicity, we only vary the block numbers in the third and fourth stages, while keeping the block numbers in the other stages unchanged. Note that the first stage corresponds to the convolutional stem. The studies with MOAT-1 meta architecture are shown in Tab. 16. In the end, we choose the layout $\{2, 2, 6, 14, 2\}$ because it has the best performance and lower parameter cost. Interestingly, our discovery echoes the layer layout proposed by CoAtNet (Dai et al., 2021). We visualize the architecture of MOAT-1 in Fig. 6.

Table 16: **Ablation studies of MOAT meta-architecture design** on ImageNet-1K, using MOAT-1 and input size 224. We control the first, second and last stages to have two blocks, and vary the block numbers of the third and fourth stages.

| number of blocks in five stages | params (M) | FLOPs (B) | top-1 acc. |
|---|---|---|---|
| (2, 2, 2, 16, 2) | 43.7 | 8.9 | 84.1 |
| (2, 2, 4, 15, 2) | 42.6 | 9.0 | 84.2 |
| (2, 2, 6, 14, 2) | 41.6 | 9.1 | 84.2 |
| (2, 2, 8, 13, 2) | 40.6 | 9.2 | 84.1 |
| (2, 2, 10, 12, 2) | 39.5 | 9.3 | 84.1 |

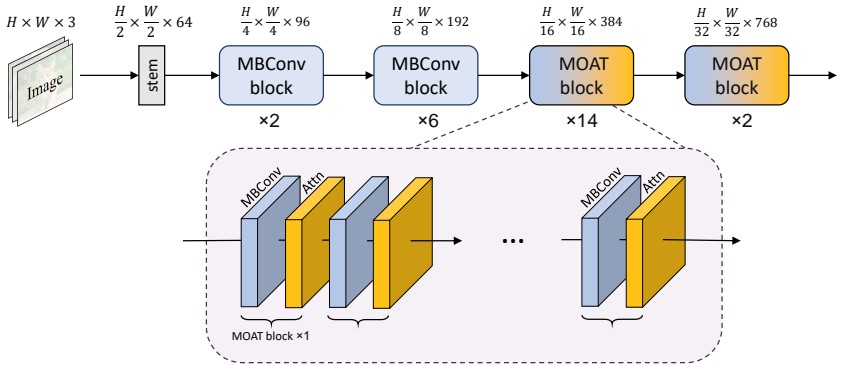

Figure 6: **Architecture of MOAT-1**, including the convolutional stem, MBConv, and MOAT blocks.

## A.7 IMAGENET TRAINING TIME, PEAK TRAINING MEMORY AND TROUGHPUT MEASUREMENTS

Table 17: **ImageNet training time** measured in hours. We use 16 TPUv4 cores for training MOAT-{0,1,2} and 32 TPUv4 cores for MOAT-3. MOAT is training efficient: for ImageNet-22k pretraining, MOAT takes no more than 2.05 days, while for ImageNet-1k pretraining, MOAT takes < 1 day.

| dataset | model | pre-training | fine-tuning | |
|---|---|---|---|---|
| | input size | $224 \times 224$ | $224 \times 224$ | $384 \times 384$ |
| ImageNet-1K | MOAT-0 | 6.5h | – | 2.8h |
| | MOAT-1 | 9.8h | – | 4.4h |
| | MOAT-2 | 13.9h | – | 6.1h |
| | MOAT-3 | 16.0h | – | 7.9h |
| | input size | $224 \times 224$ | $224 \times 224$ | $384 \times 384$ |
| ImageNet-22K | MOAT-0 | 20.1h | 0.9h | 2.5h |
| | MOAT-1 | 30.0h | 1.3h | 3.9h |
| | MOAT-2 | 42.6h | 1.8h | 5.4h |
| | MOAT-3 | 49.2h | 2.2h | 7.0h |

Table 18: **ImageNet peak training memory** of MOAT models. The input size is $224 \times 224$.

| model | training statistics | | | |
|---|---|---|---|---|
| | total batch size | num. of TPUv4 cores | batch size per core | peak memory per core (MB) |
| MOAT-0 | 4096 | 16 | 256 | 19155 |
| MOAT-1 | 4096 | 16 | 256 | 26170 |
| MOAT-2 | 4096 | 16 | 256 | 26662 |
| MOAT-3 | 4096 | 32 | 128 | 26260 |

Table 19: **ImageNet throughput measurement** of MOAT models. We re-implement MOAT with the popular "timm" (Wightman, 2019) library in PyTorch, and measure the throughput on an Nvidia V100 GPU, following the same settings as DeiT (Touvron et al., 2021a), Swin (Liu et al., 2021), and ConvNeXt (Liu et al., 2022b).

| input size | | $224 \times 224$ | | $384 \times 384$ | | $512 \times 512$ | |
|---|---|---|---|---|---|---|---|
| model | params (M) | FLOPs (B) | throughput (images/sec) | FLOPs (B) | throughput (images/sec) | FLOPs (B) | throughput (images/sec) |
| MOAT-0 | 27.8 | 5.7 | 536 | 18.2 | 155 | – | – |
| MOAT-1 | 41.6 | 9.1 | 339 | 29.6 | 91 | 58.7 | 41 |
| MOAT-2 | 73.4 | 17.2 | 209 | 54.3 | 58 | 104.6 | 27 |
| MOAT-3 | 190.0 | 44.9 | 89 | 141.2 | 23 | 271.0 | 9 |
| MOAT-4 | 483.2 | – | – | – | – | 648.5 | 4 |

## A.8 LIMITATIONS

Currently, the scaling rule of MOAT model variants are hand-designed. We, therefore, expect the architecture could be further improved by the breakthroughs in neural architecture search or network pruning (attaining faster inference speed while maintaining a similar accuracy).

