# OpenReview forum: "MOAT: Alternating Mobile Convolution and Attention Brings Strong Vision Models"
_ICLR.cc/2023/Conference — ICLR 2023 poster_

### Official Review · Reviewer_BAq9 · 2022-10-23

**Confidence:** 4
**Correctness:** 3
**Technical Novelty And Significance:** 2
**Empirical Novelty And Significance:** 3
**Recommendation:** 6

**Clarity, Quality, Novelty And Reproducibility:**

The paper is well written, there are many experiences to validate the proposed architecture. The architecture seems to be an adaptation of different already existing components. The code is promised so the results will be reproducible.



**Strength And Weaknesses:**

Strenghts:

- The paper is well written and easy to follow.
- The proposed method is simple and seems to give good results on the different settings considered.


Weakness:

-Comparison with concurent approaches: The paper states in the abstract "The research community has thus attempted to combine the strengths from both architectures. Unlike the current works that simply stack separate mobile convolution and transformer blocks, we effectively merge them into a MOAT block." However there is no analysis that compares the Moat block and other blocks proposed in the literature (CMT [1] , Yuan et al. [2], CoatNet [3])using the same architecture. In order to have a clear comparison of the Moat blocks and concurrent approaches.

[1] Guo et al., CMT: Convolutional Neural Networks Meet Vision Transformers
[2] Incorporating Convolution Designs into Visual Transformers
[3] CoAtNet: Marrying Convolution and Attention for All Data Sizes

-Comparison without EMEA:  According to Table 10 the paper use EMEA for models train on ImageNet-1k for a fairer comparison with the other approaches it would be interesting to also report the results without EMEA

- Semantic segmentation: On ADE20k the UperNet evaluation setting is more popular than the one used in the paper. Currently the comparison is quite limited. It would be interesting to have a larger comparison with the upernet setting to see how the proposed architecture performs compared to competing approaches.

- Missing metrics: There is only FLOPs and parameters that reported in the different tables. It is necessary to add the peak memory and the latency in order to better understand the different trade-offs. Indeed, MBconv are known to give good FLOPs accuracy/params-accuracy trade-offs but are not always very good for the other trade-offs.


- Overfitting evaluations:There is no evaluation of overfitting but on ImageNet the evaluation is done on the validation set which can lead to overfitting. It is important to add evaluation on ImageNet-v2 [1] to evaluate the level of overfitting.

[1] Recht et al., Do ImageNet Classifiers Generalize to ImageNet?


Other comments:
- FLOPs/ Parameters trade-off: Table 2 and Table 9 It is interesting to add all the EfficientNet-v2 models (S and M) because they have a good trade-off.


**Summary Of The Paper:**

The paper proposes a new way to incorporate MBconv in a transformers architecture. This in order to propose an efficient architecture for many tasks with good FLOPS- accuracy and parameters-accuracy trade-off.

**Summary Of The Review:**

The paper is well written and the proposed idea is interesting; however, it lacks results in order to better evaluate the significance of the results and to know the impact of the proposed method compared to the existing one in the literature.

====Post Rebuttal====
The authors address some of my concerns in particular on the experimental part. However, the proposed architecture lacks novelty and clear motivation in its design in comparison to the literature, but the experimental part is quite extensive. So I will keep my original score.

---

> ### Author Response · Authors · 2022-11-18
> **To Reviewer BAq9 (1/2)**
>
> We thank R4 for the constructive comments, and address them below.
>
> > Q1: Lacks analysis that compares the MOAT block with CMT [1], CeiT [2] and CoAtNet [3].
>
> A1: We thank the reviewer for pointing this out. We have added CMT [1] and CeiT [2] in our Related Work section. Specifically, CMT block employs 3x3 depthwise convolutions in their Local Perception Unit and the FFN (within the Transformer block), and applies strided depthwise convolution to key and value before the attention operation. CeiT inserts depthwise convolution into the feed forward network. CoAtNet [3] stacks MBConv blocks that consist of depthwise convolution, normalization and activation functions before the transformer blocks.
>
> There is no *official* TensorFlow implementation of CMT and CeiT, making it difficult for us to faithfully incorporate them to our TensorFlow code-base. Since our model is most related to CoAtNet, we carefully perform comparisons with CoAtNet below (we share the same code-base as CoAtNet and thus are able to reproduce their results).
>
> We perform experiments on both ImageNet classification and COCO object detection and instance segmentation. Please note that we use the same code-base as CoAtNet for both upstream and downstream tasks. The ImageNet results are obtained by training the model from scratch on ImageNet-1K data only with 224 input size, the COCO results are obtained by using the ImageNet-1K checkpoints as the initialization. All comparisons are ensured to be fair.
>
> As shown in the table, our MOAT performs significantly better than CoAtNet on both tasks. Specifically, MOAT-0/1 surpass CoAtNet-0/1 by 1.7/0.9% on ImageNet classification, and 5.5/6.0 box AP and 4.0/4.8 mask AP on COCO object detection and instance segmentation.
>
> | model | params (M) | top-1 acc. |
> | :--- | :---: | :---: |
> | CoAtNet-0 | 25 | 81.6 |
> | CoAtNet-1 | 42 | 83.3 |
> | MOAT-0 | 28 | 83.3 |
> | MOAT-1 | 42 | 84.2 |
>
> | model | params (M) | box AP | mask AP |
> | :--- | :---: | :---: | :---: |
> | CoAtNet-0 | 61 | 50.4 | 43.4 |
> | CoAtNet-1 | 79 | 51.7 | 44.2 |
> | MOAT-0 | 65 | 55.9 | 47.4 |
> | MOAT-1 | 79 | 57.7 | 49.0 |
>
> > Q2: Report results without EMA on ImageNet-1k.
>
> A2: We emphasize that EMA has been used in current state-of-the-art methods (and we strictly follow them), e.g., CoAtNet [3] and ConvNeXt [4]. Similar to DeiT [5] and Swin [6], we also observe that in some cases EMA improves about 0.1%, while in some cases it does not improve the performance.
>
> > Q3: Semantic segmentation on ADE20K with UPerNet setting.
>
> A3: We thank the reviewer for the suggestion. Unfortunately, there is no official TensorFlow implementation of UPerNet. On the other hand, DeepLabv3+ is a more standard semantic segmentation model, which has been well supported in both TensorFlow and PyTorch. Admittedly, UPerNet performs better than DeepLabv3+ but at the cost of more FLOPs (e.g., when using the ConvNeXt backbones, as shown in our Tab. 4 vs. the Tab. 4 in the ConvNeXt paper). We note that in Tab. 4 of our paper, we aim to demonstrate that MOAT is more effective than ConvNeXt in the downstream tasks, using the same underlying meta architecture for semantic segmentation.
>
> > Q4: Missing latency and peak memory measurements.
>
> A4: We thank the reviewer for the suggestion. We reimplement MOAT with the ‘timm’ library in PyTorch and measure the throughputs on an Nvidia V100 GPU with input size 224, following *exactly* the same way as DeiT [5], Swin [6] and ConvNeXt [4]. We also report the training peak memory on TPUv4. More results can be found in the updated paper (Appendix A.7).
>
> | model | params (M) | input size | FLOPs (B) | throughput (images/sec) |
> | :--- | :---: | :---: | :---: | :---: |
> | MOAT-0 | 27.8 | 224 | 5.7 | 536 |
> | MOAT-1 | 41.6 | 224 | 9.1 | 339 |
> | MOAT-2 | 73.4 | 224 | 17.2 | 209 |
> | MOAT-3 | 190.0 | 224 | 44.9 | 89 |
>
> | model | total batch size | num. of  TPUv4 cores  | batch size / core | peak memory / core | 1k top-1 acc. | 22k to 1k top-1 acc. |
> | :--- | :---: | :---: | :---: | :---: |:---: | :---: |
> | MOAT-0 | 4096 | 16 | 256 | 19155 | 83.3 | 83.6 |
> | MOAT-1 | 4096 | 16 | 256 | 26170 | 84.2 | 84.9 |
> | MOAT-2 | 4096 | 16 | 256 | 26662 | 84.7 | 86.0 |
> | MOAT-3 | 4096 | 32 | 128 | 26260 | 85.3 | 86.8 |

---

> > ### Author Response · Authors · 2022-11-18
> > **To Reviewer BAq9 (2/2)**
> >
> > > Q5: Evaluation on ImageNet-1K-V2.
> >
> > A5: We thank the reviewer for the suggestion. We therefore evaluate MOAT on ImageNet-1K-V2. We compare MOAT with the current state-of-the-art method SwinV2 without extra proprietary training data (i.e., only public ImageNet data).
> >
> > As shown in the table below, we make the following observations:
> > 1. MOAT does not overfit to ImageNet-1K-V1 dataset and generalizes well to ImageNet-1K-V2 dataset, as we observe a continuous performance improvement from small to large models.
> > 2. Under the fair comparison, with ImageNet-22K pretrainng and input size 384, MOAT-2/3 surpass the current state-of-the-art model SwinV2-B/L by 0.6/1.7%, respectively.
> > 3. We report an extensive evaluation, using MOAT and several input resolutions, on ImageNet-1K-V2, aiming to establish another solid baseline for the community, as we notice that most of the existing models do not report results on ImageNet-1K-V2.
> > 4. Our MOAT-4, with input size 512, achieves a new state-of-the-art performance of 81.5%, without extra proprietary training data.
> >
> > We have included more detailed comparisons with other state-of-the-art methods in the updated paper (see Appendix A.8).
> >
> > | model | params (M) | input size | training set | ImageNet-1K-V2 top-1 acc. |
> > | :--- | :---: | :---: | :---: | :---: |
> > | SwinV2-B [7] | 88 | 384 | ImageNet-22K | 78.1 |
> > | SwinV2-L [7] | 197 | 384 | ImageNet-22K | 78.3 |
> > | tiny-MOAT-0 | 3.4 | 224 | ImageNet-1K | 64.3 |
> > | tiny-MOAT-1 | 5.1 | 224 | ImageNet-1K | 67.3 |
> > | tiny-MOAT-2 | 9.8 | 224 | ImageNet-1K | 70.1 |
> > | tiny-MOAT-3 | 19.5 | 224 | ImageNet-1K | 72.1 |
> > | MOAT-0 | 27.8 | 224 | ImageNet-1K | 72.8 |
> > | MOAT-1 | 41.6 | 224 | ImageNet-1K | 74.2 |
> > | MOAT-2 | 73.4 | 224 | ImageNet-22K | 76.7 |
> > | MOAT-3 | 190.0 | 224 | ImageNet-22K | 78.4 |
> > | MOAT-2 | 73.4 | 384 | ImageNet-22K | 78.7 |
> > | MOAT-3 | 190.0 | 384 | ImageNet-22K | 80.0 |
> > | MOAT-3 | 190.0 | 512 | ImageNet-22K | 80.6 |
> > | MOAT-4 | 483.2 | 512 | ImageNet-22K | **81.5** |
> >
> > > Q6: Tab. 2 and 9 should add EfficientNet-V2 (S and M) results.
> >
> > A6: We thank the reviewer for pointing this out. We have added EfficientNet-V2 (S and M) in Tab. 2 and Tab. 9 for better comparison.
> >
> > [1] Guo, Jianyuan, et al. "Cmt: Convolutional neural networks meet vision transformers." CVPR 2022.
> >
> > [2] Yuan, Kun, et al. "Incorporating convolution designs into visual transformers." ICCV 2021.
> >
> > [3] Dai, Zihang, et al. "Coatnet: Marrying convolution and attention for all data sizes." NeurIPS 2021.
> >
> > [4] Liu, Zhuang, et al. "A convnet for the 2020s." CVPR 2022.
> >
> > [5] Touvron, Hugo, et al. "Training data-efficient image transformers & distillation through attention." ICML 2021.
> >
> > [6] Liu, Ze, et al. "Swin transformer: Hierarchical vision transformer using shifted windows." ICCV 2021.
> >
> > [7] Liu, Ze, et al. "Swin transformer v2: Scaling up capacity and resolution." CVPR 2022.

---

> ### Comment · Reviewer_BAq9 · 2022-11-30
> **Response to the rebuttal**
>
> I would like to thank the authors for their rebuttal.
>
> Here are the questions and comments I have:
>
> - Comparison with concurent approaches:
> This makes it a little more difficult to not to have results with CMT and CeiT. There is a PyTorch implementation of MoaT in the supplemental, it might have been possible to perform the comparison with the PyTorch codebase ?
> Could you share your training and validation logs to get more insight on the training?
>
> -  Comparison without EMEA:  Thanks for your answer.
>
> - Semantic segmentation:  Thanks for your answer.  There is a PyTorch implementation of MoaT in the supplemental it might have been possible to perform the comparison with the PyTorch codebase ?
>
> - Missing metrics: Could you report the results for others architecture like ConvNexT, DeiT or EfficientNet with the same setting in order to have some comparison ? Currently it is not possible to conclude with these results.
>
> - Overfitting evaluations: Thanks for the results, could you add all the MoaT models from Table 9 ?

---

> > ### Author Response · Authors · 2022-12-05
> > **Second rebuttal to reviewer BAq9 (1/3)**
> >
> > We thank the reviewer for the additional comments. We will address them one-by-one below. Before that, we would like to make the following suggestions:
> >
> > 1. We kindly ask the reviewer to be more inclusive, and considerate of the different infrastructures between TPU-TensorFlow and GPU-PyTorch. Our infrastructure mainly supports the TPU-TensorFlow framework, making it infeasible for us to perform large-scale GPU-PyTorch training, and we can only use GPU-PyTorch for small-scale experiments, such as evaluation jobs. We have tried our best to ensure fairness in all the reported experiments.
> >
> > 2. We kindly request the reviewer to provide more detailed instructions, and cautiously exercise their reviewer duty when requesting for more experiments, as we have been carefully conducting each experiment (both in the paper and rebuttal) and each experiment usually takes us hours or even days to ensure correctness.
> >
> > We now address the reviewer’s comments one-by-one in the following posts.

---

> > > ### Author Response · Authors · 2022-12-05
> > > **Second rebuttal to reviewer BAq9 (2/3)**
> > >
> > > > Q1: Comparison with CMT and CeiT.
> > >
> > > A1: We have carefully studied CMT and CeiT over the last week. We systematically compare CMT, CeiT and MOAT from the perspectives of (1) macro-level, and (2) micro-level designs.
> > >
> > > ***Macro-level comparison***: At the macro-level comparison, we take their proposed meta architecture and block design from each paper, and compare them below, where the results of CMT are obtained from their Tab. 2, and CeiT are from their Tab. 4. As shown in the tables below, we observe the following things:
> > >
> > > 1. MOAT is slightly more parameter-efficient than CMT. When using similar parameters, MOAT performs slightly better, except MOAT-0 vs. CMT-S. Particularly, when using input size 288x288, MOAT-2 (73.4M) reaches 85.2% top-1 accuracy, outperforming CMT-L (74.7M) by 0.4%. On the other hand, CMT is more FLOPs-efficient than MOAT, which is attributed to their design of lightweight MHSA, where an extra strided depthwise convolution is applied to the key and value.
> > >
> > > 2. MOAT performs significantly better than CeiT across different input sizes and model sizes. We note that CeiT adopts a ViT-like isotropic meta architecture, and thus uses fewer FLOPs.
> > >
> > > | model | input size | params (M) | FLOPs (B) | top-1 acc. |
> > > | :--- | :---: | :---: | :---: |:---: |
> > > | CMT-Ti | 160 | 9.5 | 0.6 | 79.1 |
> > > | CMT-S | 224 | 25.1 | 4.0 | 83.5 |
> > > | CMT-B | 256 | 45.7 | 9.3 | 84.5 |
> > > | CMT-L | 288 | 74.7 | 19.5 | 84.8 |
> > > | tiny-MOAT-2 | 160 | 9.8 | 1.1 | 79.5 |
> > > | MOAT-0 | 224 | 27.8 | 5.7 | 83.3 |
> > > | MOAT-1 | 256 | 41.6 | 12.1 | 84.8 |
> > > | MOAT-2 | 288 | 73.4 | 29.2 | 85.2 |
> > >
> > > | model | input size | params (M) | FLOPs (B) | top-1 acc. |
> > > | :--- | :---: | :---: | :---: |:---: |
> > > | CeiT-T | 224 | 6.4 | 1.2 | 76.4 |
> > > | CeiT-S | 224 | 24.2 | 4.5 | 82.0 |
> > > | CeiT-T_384 | 384 | 6.4 | 3.6 | 78.8 |
> > > | CeiT-S_384 | 384 | 24.2 | 12.9 | 83.3 |
> > > | tiny-MOAT-1 | 224 | 5.1 | 1.2 | 78.3 |
> > > | MOAT-0 | 224 | 27.8 | 5.7 | 83.3 |
> > > | tiny-MOAT-1 | 384 | 5.1 | 4.2 | 80.9 |
> > > | MOAT-0 | 384 | 27.8 | 18.2 | 84.6 |
> > >
> > > ***Micro-level comparison***: At the micro-level comparison, we adopt the same MOAT-0 meta architecture, but use different blocks in the last two stages. We carefully implement the CMT block and CeiT block in our TPU-TensorFlow framework by **strictly** following their official open-source implementation [CMT](https://github.com/ggjy/CMT.pytorch) and [CeiT](https://github.com/coeusguo/ceit). All the comparisons are ensured to be fair.
> > >
> > > As shown in the table below, our MOAT block demonstrates the best performance. Interestingly, using CeiT blocks shows comparable performance, echoing the finding from the community that using multi-scale meta architecture is beneficial (original CeiT adopts a ViT-like isotropic meta architecture). We note that this experiment does not necessarily indicate that CMT block is inferior to CeiT or MOAT blocks. It simply shows that the CMT block is not suitable for our MOAT meta architecture (CMT-S meta architecture adopts a deeper structure than MOAT-0). We think that the CMT paper is a great work and it contains multiple novel designs. However, exploring the best meta architectures for those block types is way beyond the scope of this work and this rebuttal.
> > >
> > > | MOAT-0 w/ block type | params (M) | FLOPs (B) | top-1 acc. |
> > > | :--- | :---: | :---: | :---: |
> > > | CMT block | 29.4 | 5.0 | 82.1 |
> > > | CeiT block | 28.3 | 5.4 | 83.0 |
> > > | MOAT block | 27.8 | 5.7 | 83.3 |
> > >
> > > We agree with the reviewer that both CMT and CeiT are important pioneering works on combining convolutions and transformers. We will include the above comparisons in the final version. Considering the limited rebuttal time, we hope the reviewer is satisfied with the results.
> > >
> > > Additionally, we are happy to provide the requested training and validation logs after we get more detailed instructions. Particularly, what exact experiments and exact statistics is the reviewer requesting for? Also, how can we upload those logs to OpenReview (the supplementary material upload link is turned off now)? Most importantly, what insights do the reviewer want to get from checking the logs? We would also like to get the agreement (and instructions to do so) from the **AC** to proceed with this request, as we believe this is not a common request.

---

> > > > ### Author Response · Authors · 2022-12-05
> > > > **Second rebuttal to reviewer BAq9 (3/3)**
> > > >
> > > > > Q2: Semantic Segmentation with UPerNet.
> > > >
> > > > A2: As mentioned in our previous rebuttal, there is no official TensorFlow implementation of UPerNet (if there is, we would definitely use it since it shows better performance than the used DeepLabv3+ meta architecture). It is infeasible for us to repeat all our semantic segmentation experiments in GPU-PyTorch, given our limited infrastructure and limited rebuttal time. We emphasize again that in Tab. 4 of our paper, we aim to demonstrate that MOAT is more effective than ConvNeXt in the downstream semantic segmentation task, using the same meta architecture DeepLabv3+. If the reviewer thinks it is improper to report this result (and concurred by the **AC**), we are willing to remove it from the main paper.
> > > >
> > > > > Q3: Report throughput and memory for other architectures like ConvNexT, DeiT or EfficientNet.
> > > >
> > > > A3: We would like to emphasize that ***not all the published papers*** include the requested metrics, such as throughput and training memory, as those statistics heavily depend on the underlying hardware and software behind the scene, making it hard to fairly compare between methods. We agree with the reviewer that it is informative to include those statistics, and we are willing to build a baseline for the community as well. However, it is ***beyond the scope of our work and this rebuttal*** to reproduce the missing statistics of **all** the published works. Additionally, we emphasize that we **never** claim *our method to be the fastest in terms of throughput or the most memory-efficient method than the other existing works*. Nevertheless, we have measured the throughput of the **newly** mentioned architectures: ConvNeXt, DeiT, and EfficientNet, using the same setting with `timm’ (GPU-PyTorch) on a V100 GPU (Float32).
> > > >
> > > > We have listed our memory statistics in the last rebuttal. For comparison, the ConvNeXt paper reported that the *isotropic* ConvNeXt-L requires 20.4 GB memory on V100 GPUs with 32 per-GPU batch size (the ConvNeXt paper even did not have a complete report of their training memory). We note that unfortunately, our TPU-TensorFlow ***ImageNet*** code-base does not support ConvNeXt, DeiT, and EfficientNet, which may take more time than the rebuttal period to reproduce their results and correctly measure their training peak memory in our TPU-TensorFlow framework. If the reviewer insists on this statistics for **other** architectures (which however is **not** the main focus of this work), we would like to invite the **AC** to mediate the dispute.
> > > >
> > > > | model | params (M) | input size | FLOPs (B) | throughput (images/sec) | top-1 acc. |
> > > > | :--- | :---: | :---: | :---: |:---: | :---: |
> > > > | EfficientNetV2-S | 21.5 | 384 | 8.8 | 359 | 83.9 |
> > > > | EfficientNetV2-M | 54.1 | 480 | 24 | 116 | 85.1 |
> > > > | DeiT-S | 22.1 | 224 | 4.6 | 938 | 79.8 |
> > > > | DeiT-B | 86.6 | 224 | 17.6 | 312 | 81.8 |
> > > > | ConvNeXt-T | 28.6 | 224 | 4.5 | 759 | 82.1 |
> > > > | ConvNeXt-S | 50.2 | 224 | 8.7 | 446 | 83.1 |
> > > > | ConvNeXt-B | 88.6 | 224 | 15.4 | 292 | 83.8 |
> > > > | ConvNeXt-L | 197.8 | 224 | 34.4 | 147 | 84.3 |
> > > > | MOAT-0 | 27.8 | 224 | 5.7 | 536 | 83.3 |
> > > > | MOAT-1 | 41.6 | 224 | 9.1 | 339 | 84.2 |
> > > > | MOAT-2 | 73.4 | 224 | 17.2 | 209 | 84.7 |
> > > > | MOAT-3 | 190.0 | 224 | 44.9 | 89 | 85.3 |
> > > >
> > > > > Q4: Add all MOAT variants in Tab. 9 for ImageNet-V2 evaluation.
> > > >
> > > > A4: Below, please find the ImageNet-V2 results for all the MOT variants (from Tab. 9). We note again that ***only few published papers*** include the ImageNet-V2 results, making it hard for us to compare between methods. Nevertheless, we agree with the reviewer that it is informative to include the ImageNet-V2 results, and we are willing to build a baseline for the community as well.
> > > >
> > > > | model | params (M) | input size | ImageNet-1k pretraining | ImageNet-22k pretraining |
> > > > | :--- | :---: | :---: | :---: | :---: |
> > > > | SwinV2-B [7] | 88 | 384 | -- | 78.1 |
> > > > | SwinV2-L [7] | 197 | 384 | -- | 78.3 |
> > > > | MOAT-0 | 27.8 | 224 | 72.8 | 74.1 |
> > > > | MOAT-1 | 41.6 | 224 | 74.2 | 75.8 |
> > > > | MOAT-2 | 73.4 | 224 | 74.3 | 76.7 |
> > > > | MOAT-3 | 190.0 | 224 | 75.5 | 78.4 |
> > > > | MOAT-0 | 27.8 | 384 | 74.5 | 76.4 |
> > > > | MOAT-1 | 41.6 | 384 | 76.2 | 78.1 |
> > > > | MOAT-2 | 73.4 | 384 | 76.5 | 78.7 |
> > > > | MOAT-3 | 190.0 | 384 | 77.5 | 80.0 |
> > > > | MOAT-1 | 41.6 | 512 | 76.8 | 78.4 |
> > > > | MOAT-2 | 73.4 | 512 | 77.1 | 79.3 |
> > > > | MOAT-3 | 190.0 | 512 | 77.8 | 80.6 |
> > > > | MOAT-4 | 483.2 | 512 | -- | **81.5** |

---

> > > > > ### Comment · Reviewer_BAq9 · 2022-12-05
> > > > > **Response to the rebuttal**
> > > > >
> > > > > First of all, I would like to thank the authors for their answer this is really helpful.
> > > > >
> > > > > I understand completely the specificities of each infrastructures as I found some additional PyTorch code in supplemental I was just wondering if the authors had any comparison with this implementation as they mentioned *"There is no official TensorFlow implementation of CMT and CeiT, making it difficult for us to faithfully incorporate them to our TensorFlow code-base."*  and *"Unfortunately, there is no official TensorFlow implementation of UPerNet"*. I did not require any additional experiments.
> > > > >
> > > > > - Comparison with concurrent: Thanks for the detailed comparison. I ask if the authors could share their training logs in order to have a better estimate of the convergence curves and the training speed. Indeed, this is emphasised in CeIT so this was to help me get more insight into the comparison with this approach.
> > > > >
> > > > > - Semantic segmentation: Thanks for your answer.
> > > > >
> > > > > - Missing metrics: Thanks for your answer. I agree with the author *"we never claim our method to be the fastest in terms of throughput or the most memory-efficient method than the other existing work"*.  However, it reinforces the submission to have a comparison with the different metrics to better understand the different trade-offs of the proposed method.
> > > > >
> > > > > - Overfitting evaluations: Thanks for your answer.

---

> > > > > > ### Author Response · Authors · 2022-12-05
> > > > > > **Thanks to reviewer BAq9**
> > > > > >
> > > > > > We sincerely thank the reviewer for the clarification, feedback, and reviews.
> > > > > >
> > > > > > If the reviewer has more questions, please kindly let us know and we are happy to discuss.
> > > > > >
> > > > > > Finally, as suggested, we will incorporate the rebuttal results to the final version.

---

### Official Review · Reviewer_W4v5 · 2022-10-24

**Confidence:** 5
**Correctness:** 4
**Technical Novelty And Significance:** 2
**Empirical Novelty And Significance:** 2
**Recommendation:** 6

**Clarity, Quality, Novelty And Reproducibility:**

**Clarity**

The overall paper is well-written. However, the authors are bypassing the thorough discussions with the recent MaxViT paper. I encourage the authors to present the common observations, distinctions, and practical advantages over the MaxViT.

**Quality**

The paper is well structured. Also, the extensive experiments and strong results are presented.

**Novelty**

The technical novelty is quite limited, considering the MaxViT paper.

**Reproducibility**

The authors open-sourced the code bases. This will help the community to easily reproduce the results.

**Details Of Ethics Concerns:**

I see no ethical issues in this paper.

**Strength And Weaknesses:**

**Strength**

*+* The paper conducts extensive experiments. Especially having dense reference points on the broad regime of model sizes shows the strong scaling effect of the model and can allow the follow-up studies to compare it more thoroughly. I appreciate the authors for doing this.

*+* The open-sourced code bases/checkpoints can provide broad visibility and high reproducibility to the community.

*+* The MOAT achieves strong results in various vision benchmarks.

**Weakness**

*-* Given the recently presented MaxVIT, my main concern with this paper is ***limited architectural innovations***. The major observations and the resulting architectural changes are 1) MBconv + self-attention (to equip with the translation equivariance and locality) and 2) strided dwconv instead of extra patch-embedding layers (to equip with the hierarchy). However, both are already presented in the MaxViT. The remaining difference is the existence of SE in the MBConv block, which is a minor change (In fact, there is a recent trial of this in MaxViT by the 3rd group in https://github.com/rwightman/pytorch-image-models, i.e., MaxViT with ConvNeXt block, and works well).

*-* Moreover, unlike MaxViT, which uses linear complexity self-attention, the MOAT directly adopts the quadratic complexity global self-attention. This makes the authors introduce inhomogeneous block configurations (i.e., using MBconv blocks at the first two stages) to reduce overeheads and perform post-hoc engineering (i.e., using window attention and setting the proper window size) to apply the model to high-resolution inputs. These reduce actual ***algorithmic simplicity*** overall.

**Summary Of The Paper:**

The paper presents a new transformer-based architecture for vision tasks, namely MOAT.
The key idea is to combine the convolutional priors, locality, and hierarchy into the pure transformer block design (self-attention, then MLP) and remove redundant operations.
First, the locality is achieved by replacing the MLP block with a mobile convolution block (MBconv). In practice, the authors empirically verified that placing the MBconv before self-attention provides the best result.
Second, the hierarchy is achieved by introducing strided depth-wise convolution in the first block in each stage, and it is efficient compared to having additional patch embedding layers.
Finally, the authors sweep the best macro-level block configurations and found that using MBconv blocks and MOAT blocks at the first and last two stages strikes the best efficiency-accuracy trade off.
The final model MOAT provides good basis for the model scaling (both up and downward) and achieves strong results in various vision tasks.




**Summary Of The Review:**

My main concern with this paper is the limited technical contribution. Given the MaxViT, the claimed technical contributions are not new. The MOAT can be regarded as an incremental variant of MaxViT. It works slightly better with proper post-hoc engineering (e.g., introduce pure MBconv blocks in early stages, use global attention and convert it to the window attention if test-time input resolution is large, etc.).

Overall, I appreciate extensive experiments and engineering efforts. The empirical results, code, and model checkpoints will help the practitioners. However, this cannot be the basis for a new ICLR conference paper. I am willing to hear the author's view on comparing MOAT with the MaxViT in various aspects, including experiment-level comparisons in a broad model-size spectrum (e.g., in which model regime the MOAT outperforms/underperforms the MaxViT?) and thorough descriptions of key differences and their resulting practical dis-/advantages (e.g., what is the core that makes MOAT practically strong compared to MaxViT?).

---

> ### Author Response · Authors · 2022-11-18
> **To Reviewer W4v5 (1/2)**
>
> We thank R3 for the constructive comments, and address them below.
>
> > Q: Comparison with MaxViT.
>
> A: We thank R3 for raising the concern with MaxViT. We will address it from the perspectives of  ***limited architectural innovations*** and ***algorithmic simplicity***. Before that, we would like to emphasize that our work is ***concurrent*** with MaxViT, as the work is done independently among research groups. It is unfair to consider our work as an incremental follow-up of MaxViT. Most of our experiments were finished in the middle of May, and technically MaxViT ECCV 2022 camera-ready is publicly viewable after 10/14/2022, after the ICLR submission deadline. We are happy to provide concrete evidence (e.g., a snapshot of our previous submission in May), if the reviewer insists and that will not violate the ICLR review policy (confirmed by the **AC**). Nevertheless, we have carefully looked into MaxViT during the rebuttal period, and please see the details below.
>
> ***limited architectural innovations***: We first note that MaxViT does not ***integrate*** MBConv into the Transformer block, and they do not discover the effectiveness of performing downsampling within the MBConv (even though they use it), as we do and have discussed in the paper. Instead, a MaxViT block contains MBConv, Block Attention, and Grid Attention, where they approximate the global attention with the window and grid attention, allowing the *claimed* linear complexity on arbitrary input resolution. Unfortunately, MaxViT design contains a critical flaw: the MaxViT model is exceedingly more complicated than our MOAT model. Specifically,
>
> 1. A MaxViT block actually contains **five** *residual blocks* (MOAT block has only two), making the network unnecessarily deep and inefficient. To be concrete, the MaxViT-B (120M parameters) variant contains 120 (=24x5) residual blocks, and uses 40 (=8x5) residual blocks in the high-resolution features (stride 4 and stride 8 totally). On the contrary, MOAT-2 (73M parameters) contains only 40 (=8+16x2) residual blocks (2+6=8 residual blocks for high-resolution features), and even MOAT-3 (190M) has only 74 (=14+30x2) residual blocks (2+12=14 residual blocks for high-resolution features).
>
> 2. A MaxViT block requires multiple *reshape* and *transpose* operations, as shown in their [open-source code](https://github.com/google-research/maxvit/blob/main/maxvit/models/maxvit.py#L683). The reshape and transpose operations are hardware-unfriendly, making their models actually ***not that efficient*** as claimed.
>
> To empirically prove our claims, we compare MOAT with MaxViT in the setting of training from scratch on ImageNet-1K with 224 input size. We use the third party library ‘timm’ (PyTorch) to measure the throughputs for both MOAT and MaxViT on a V100 GPU following the *same* setting as Swin and ConvNeXt. All the comparisons are ensured to be fair.
>
> As shown in the table, our MOAT performs comparably to MaxViT at similar params and FLOPs (MOAT-1/2 are slightly more efficient than MaxViT-S/B). However, **all** our throughputs are better than MaxViT, whose linear complexity is actually sacrificed by (1) their unnecessary deep structure, and (2) hardware-unfriendly reshape and transpose operations.
>
> Additionally, the deep structure design of MaxViT makes it hard to scale up for huge models. MaxViT stops increasing the network depth after MaxViT-B, and their model performance starts to saturate after MaxViT-B as shown in their paper’s Fig. 1. (b) and Fig. 4. (a) when using only public ImageNet data. On the other hand, our best model MOAT-4 (483M params) achieves 89.1% top-1 accuracy, surpassing the MaxViT-XL (475M) by **0.4%**, which is significant as the performance is approaching 90%.
>
> | model | params (M) | FLOPs (B) | **throughput (images/sec)** | top-1 acc. |
> | :--- | :---: | :---: | :---: |:---: |
> | MaxViT-T | 31 | 5.6 | 409 | 83.6 |
> | MaxViT-S | 69 | 11.7 | 255 | 84.5 |
> | MaxViT-B | 120 | 23.4 | 131 | 85.0 |
> | MaxViT-L | 212 | 43.9 | 87 | 85.2 |
> | MOAT-0 | 27.8 | 5.7 | 536 | 83.3 |
> | MOAT-1 | 41.6 | 9.1 | 339 | 84.2 |
> | MOAT-2 | 73.4 | 17.2 | 209 | 84.7 |
> | MOAT-3 | 190.0 | 44.9 | 89 | 85.3 |
>
> Finally, as mentioned by the reviewer, we take a careful look at the ***third party*** *advanced* implementation of MaxViT in the timm library.
>
> We note that the advanced improvement by timm, *maxxvit_rmlp_nano_rw_256*, contains multiple tricks, including using the ConvNeXt block, RelPosMLP, and also a series of changes as shown in the [timm open-source code](https://github.com/rwightman/pytorch-image-models/blob/main/timm/models/maxxvit.py#L251). Nevertheless, our tiny-MOAT-3 still outperforms it and has higher throughput. We also note that those tricks can improve CoAtNet-1 and CoAtNet-2 by 0.3% and 0.5%, respectively (called *coatnet_1_rw_224* and *coatnet_rmlp_2_rw_224* in timm), and thus could be also potentially incorporated into MOAT.

---

> > ### Author Response · Authors · 2022-11-18
> > **To Reviewer W4v5 (2/2)**
> >
> > | model | params (M) | throughput (images/sec) | top-a acc. |
> > | :--- | :---: | :---: | :---: |
> > | tiny-MOAT-3 | 19.5 | 488 | 83.3 |
> > | maxxvit_rmlp_nano_rw_256 | 16.8 | 448 | 83.0 |
> >
> > ***algorithmic simplicity***: We respectfully disagree with the reviewer for *algorithmic simplicity*, but instead embrace the *transferability* and *generalizability*. Our work is mostly influenced by CoAtNet, which conducted a systematic study of the layer layout of MBConv and Transformer blocks, and reached the conclusion that a better generalizability is achieved by placing MBConv in the first two stages of the network. Our empirical findings echo their principle, and thus our final macro-level design of MOAT-1 is similar to CoAtNet-1.
> >
> > To empirically prove that MaxViT has worse *transferability* (even though they have “algorithmic simplicity” in the network design except the convolutional stem), we conduct experiments on the standard downstream benchmark, COCO object detection and instance segmentation, where MaxViT does not demonstrate the outstanding performance as they achieve on ImageNet (their deep structure design may overfit to ImageNet).
> >
> > We use the same code-base as MaxViT, i.e., the Cascade Mask-RCNN framework in the open-source TF-Vision Model Garden. We additionally register the operation “tf.einsum” when calculating the FLOPs, which TF-Vision Model Garden forgets to do. All the backbones are pretrained on ImageNet-1K dataset only. We follow the same input size setting 896x896 as MaxViT. All the comparisons are ensured to be fair.
> >
> > As shown in the table, our MOAT demonstrates a much better transferability and generalizability (from ImageNet to COCO) on the downstream tasks. In addition to being more efficient in terms of model parameters and FLOPs, our MOAT-0/1/2 ***significantly*** surpass MaxViT-T/B/L by 3.0/4.0/4.0 box AP and 2.4/3.2/3.0 mask AP. The performance improvement proves the superiority of our MOAT design that carefully merges mobile convolution block and transformer block into one block, instead of naively stacking them as MaxViT does.
> >
> > | model | input size | params (M) | FLOPs (B) | box AP | mask AP |
> > | :--- | :---: | :---: | :---: |:---: |:---: |
> > | MaxViT-T | 896 | 68.6 | 521.4 | 52.1 | 44.6 |
> > | MaxViT-S | 896 | 106.8 | 619.2 | 53.1 | 45.4 |
> > | MaxViT-B | 896 | 157.4 | 817.8 | 53.4 | 45.7 |
> > | MOAT-0 | 896 | 64.9 | 525.2 | 55.1 | 47.0 |
> > | MOAT-1 | 896 | 78.6 | 579.5 | 57.1 | 48.6 |
> > | MOAT-2 | 896 | 110.3 | 710.1 | 57.4 | 48.7 |
> >
> > Finally, we note that we do not perform any *post-hoc engineering* (referred by the reviewer as “using window attention and setting the proper window size”) to apply the model to high-resolution inputs. When applying MOAT to high-resolution inputs, we **simply** use the window attention with size 14, the same as the “window size” used in ImageNet pretraining (input 224x224 has feature map size 14x14 at stride 16). Theoretically, MaxViT also has the hyper-parameters to engineer their window and grid sizes.

---

> > > ### Comment · Reviewer_W4v5 · 2022-11-30
> > > **Rebuttal Response.**
> > >
> > > Thanks for the hard work during the rebuttal.
> > > The authors tried to clarify most of my concerns and thus I will raise my score.
> > >
> > > The remaining concern is on the limited novelty.
> > > The high-level concept of using depth-wise convolution first and then self-attention is not new (The first version of the MaxViT was on the arXiv far earlier than the ECCV.).
> > > The claims/analyses on the difference between the integration (MOAT) and addition (MaxViT) of the depth-wise convolution are post-hoc,
> > > I don't see any initial motivation for designing in this way in the main paper.
> > > I encourage the authors to include the current discussions in the final version.

---

> > > > ### Author Response · Authors · 2022-12-05
> > > > **Thanks to reviewer W4v5**
> > > >
> > > > We sincerely thank the reviewer for the review and feedback.
> > > >
> > > > As suggested, we will summarize the discussion about MOAT and MaxViT in the final version.

---

### Official Review · Reviewer_bFXs · 2022-10-24

**Confidence:** 4
**Correctness:** 4
**Technical Novelty And Significance:** 2
**Empirical Novelty And Significance:** 3
**Recommendation:** 6

**Clarity, Quality, Novelty And Reproducibility:**

There are a few places that the text is unclear, which leads it is difficult to reproduce as the codes are not provided. E.g.
1. at the first observation, if I understand correctly, the transformer block is not just MLP + Attention anymore, the authors added  depthwise convolution, normalization and activation, but I am not able to see how they are added or does the authors simply mean changing the MLP to MBConv?

**Strength And Weaknesses:**

Strength
1. The authors demonstrate the efficiency and effectiveness on the proposed MOAT module.
2. The ablation studies are comprehensive, especially the order of MBConv and Attention block.

Weaknesses
1. The third observation seems to be found by other works already, e.g. PiT and RegionViT both use depthwise convolution for the donwsampling and they found it is effective.

**Summary Of The Paper:**

This paper proposed a new building block, which combined MBConv and Attention, for neural network design and the authors show its efficiency and effectiveness on many visual tasks.

**Summary Of The Review:**

Please see my comments in Section 1, 2 and 3.

And I have other questions:
1. In the abstract, authors mentioned that the generalization of transformer is worse than ConvNet, but is there any experiment to support this argument?

2. In Table 8, this ablation study focus on the effects of downsampling layer, why the block composition is also varied in this case?

3. Under tiny-MOAT setting (Table 5), what is the performance of tiny-MOAT if we would like to match the FLOPs to 0.05B from MobileFormer?

---

> ### Author Response · Authors · 2022-11-18
> **To Reviewer bFXs**
>
> We thank R2 for the constructive comments, and address them below.
>
> > Q1: PiT and RegionViT both use depthwise convolution for the downsampling and they found it is effective.
>
> A1: We thank the reviewer for pointing this out. In fact, our downsampling scheme is very different from the one used in PiT and RegionViT, which apply an extra strided 3x3 depthwise convolution for downsampling. Their scheme, denoted as “StridedDepthConv + Attn + MLP” in the table below, is actually similar to Swin and ConvNeXt by replacing the PatchEmbedding (strided 2x2 convolution) with the strided 3x3 depthwise convolution. The StridedDepthConv downsampling scheme leads to 0.2% worse performance than the PatchEembedding scheme, but uses slightly fewer parameters and FLOPs. On the other hand, MOAT performs the strided depthwise convolution **in the** MBConv block, which additionally enriches the model representation capacity by using the 1x1 convolutions (for channel expansion and projection) and extra normalization/activation in-between convolutions. We have updated the Tab. 8 in the paper to include the downsampling scheme used by PiT and RegionViT.
>
> | block composition | downsampling type | params (M) | FLOPs (B) | top-1 acc. |
> | :--- | :---: | :---: | :---: |:---: |
> | AveragePooling + Attn + MLP | CoAtNet | 28.0 | 5.4 | 82.6 |
> | PatchEmbedding + Attn + MLP | Swin, ConvNeXt | 30.2 | 5.6 | 82.8 |
> | StridedDepthConv + Attn + MLP | PiT, RegionViT | 28.8 | 5.5 | 82.6 |
> | MBConv + Attn (ours) | MOAT | 27.8 | 5.7 | 83.3 |
>
> > Q2: How to reproduce MOAT? E.g. How are the depthwise convolution, normalization and activation added? Do authors mean replacing MLP to MBConv?
>
> A2: We have uploaded our anonymous code (in both TensorFlow and PyTorch) as the supplementary material for reproducing MOAT. Please refer to it for the layout of convolution, normalization and activations, which is the same as illustrated in Fig. 1 (c) in the paper.
>
> To clarify a bit more, the transformer block is denoted as “Attention + MLP”, while MOAT is “MBConv + Attention”, where the “MBConv” is  a combination of convolution layers with normalization and activation properly added (see Fig. 1 (a), but *without* the SE module), while the “MLP” is two fully connected layers with one activation layer in between.
>
> > Q3: Is there any experiment to support the argument in the abstract that the generalization of transformers is worse than ConvNet?
>
> A3: We thank the reviewer for the question. The generalization of transformers is worse than ConvNets are demonstrated in the original ViT [A] and CoAtNet [B] papers, where ViT demonstrates that vision transformers requires a large-scale proprietary dataset (e.g., JFT-300M) to learn the inductive bias built in ConvNets, and CoAtNet studies a principle way to stack convolution layers and attention layers to improve the generalization. It is not considered as our contribution in the paper (and we never claimed that as our novelty either). If the reviewer thinks it is improper to mention that in the abstract, we are happy to rephrase it.
>
> > Q4: Table 8 is the ablation study for effects of downsampling layer, why is the block composition also varied?
>
> A4: In Table 8, the block composition is to illustrate how the different types of downsampling are achieved. We clearly specify that the downsampling scheme used by CoAtNet is applying an Average Pooling before the transformer block (denoted as *AveragePooling* + Attn + MLP), while Swin adopts a Patch Embedding layer before the transformer block (denoted as *PatchEmbedding* + Attn + MLP). On the other hand, MOAT downsamples the features by the MBConv before the attention (denoted as MBConv + Attn).
>
> > Q5: In Table 5, what is the performance of tiny-MOAT if we would like to match the FLOPs to 0.05B from MobileFormer?
>
> A5: We thank the reviewer for the interesting question. We notice that MobileFormer has small FLOPs, since they perform the attention operation on *a very few* latent variables (e.g., 6 in their experiments) instead of on the original pixel space. This is an interesting direction to explore for reducing the model complexity.
>
> [1] Heo, Byeongho, et al. "Rethinking spatial dimensions of vision transformers." ICCV 2021.
>
> [2] Chen, Chun-Fu, et al. "Regionvit: Regional-to-local attention for vision transformers." ICLR 2022.
>
> [A] Dosovitskiy, Alexey, et al. "An image is worth 16x16 words: Transformers for image recognition at scale." ICLR 2021.
>
> [B] Dai, Zihang, et al. "Coatnet: Marrying convolution and attention for all data sizes." NeurIPS 2021.

---

> > ### Comment · Reviewer_bFXs · 2022-11-30
> > **Thanks for the response**
> >
> > The authors addressed my concerns and I keep my rating. Thanks.

---

> > > ### Author Response · Authors · 2022-12-05
> > > **Thanks to reviewer bFXs**
> > >
> > > We sincerely thank the reviewer for the review and feedback.

---

### Official Review · Reviewer_GsU5 · 2022-10-26

**Confidence:** 3
**Correctness:** 3
**Technical Novelty And Significance:** 2
**Empirical Novelty And Significance:** 3
**Recommendation:** 6

**Clarity, Quality, Novelty And Reproducibility:**

Clarity: It's easy to follow. It presents its motivation based on two key observations which make sense to me.
Quality: Writing is good and experiments are sufficient.
Novelty: Novelty is a little weak for me. It is more technical by combining MobileConvet and Transformer.
Reproducibility: I think reproducibility is fine.

**Strength And Weaknesses:**

Strength: Paper writing is good and easy to follow; Idea is simple and effective; Experiments are sufficient.

Weaknesses: The paper is more like a technical report. The novelty is weak by combing MobileConvet and Transformer.

**Summary Of The Paper:**

This paper presents a family of neural networks called MOAT, which combines Mobile ConvNet and Transformer. It studies how to build effective networks based on two observations from Mobile ConvNet and Transformer. The new proposed networks achieve good performance on different tasks.

**Summary Of The Review:**

Refer to the above sections.

---

> ### Author Response · Authors · 2022-11-18
> **To Reviewer GsU5**
>
> We thank R1 for the review, and we address the concerns below.
>
> > Q1: The novelty is limited by combining mobile ConvNet and transformers.
>
> A1: We emphasize that our MOAT block design is not a mere naive combination of mobile ConvNet and transformers. Instead, MOAT block **seamlessly** integrates MBConv and Transformer into **one** block. The MBConv in a MOAT block effectively brings three advantages: (1) it enhances the network representation capacity over the original FFN block, (2) it produces better downsampled features than the standard patch-embedding (in Swin or ConvNeXt) and average-pooling (in CoAtNet) operations, and (3) it allows us to seamlessly apply MOAT to downstream tasks without any other specific design, e.g., the window-shifting mechanism as in Swin. We have carefully ablated the model design and shared the design insights in the paper. To the best of our knowledge, none of the existing works have studied the *micro-level* design to combine mobile ConvNet and Transformers.
>
> As a result, our MOAT demonstrates two important properties: **scalability** and **transferability** that none of the existing works have considered or achieved.
>
> **scalability**: Our MOAT could be easily scaled up or down, and demonstrate strong scalability and performance across *all* model sizes. Specifically, for the mobile models, tiny-MOAT-1/2/3 surpass MobileFormer and MobileViT-V2 by 5.5/4.3/3.4% and 1.1/1.3/2.1%, respectively. For the large model, MOAT-4 attains 89.1% accuracy, achieving the new state-of-the-art performance and surpassing CoAtNet by 0.5% accuracy. All the other methods require specific designs to different model sizes, while MOAT shows the strongest performance by *simply* being scaled down and up.
>
> **transferability**: Our MOAT demonstrates solid transferability and performance on many core computer vision downstream tasks, including COCO object detection and instance segmentation, ADE20K semantic segmentation, and COCO panoptic segmentation. Our design is seamlessly applied to the downstream tasks without using any extra specific designs. To be concrete, we take CoAtNet (most related to our work), which treats mobile convolution and transformer blocks separately and stacks them, for a comparison on COCO detection (using the same setting). As shown in the table, CoAtNet does not transfer well to the downstream task. Our MOAT-0 and MOAT-1 demonstrate **5.5%** AP and **6.0%** box AP improvement over CoAtNet-0 and CoAtNet-1, respectively.
>
> | model | params (M) | box AP | mask AP |
> | :--- | :---: | :---: | :---: |
> | CoAtNet-0 | 61 | 50.4 | 43.4 |
> | CoAtNet-1 | 79 | 51.7 | 44.2 |
> | MOAT-0 | 65 | 55.9 | 47.4 |
> | MOAT-1 | 79 | 57.7 | 49.0 |
>
> > Q2: The paper is more like a technical report.
>
> A2: We respectfully disagree with the reviewer, and kindly solicit constructive feedback from the reviewer. As shown in **A1** above, to the best of our knowledge, none of the existing works have carefully studied the combination of mobile ConvNet and Transformer at the *micro-level* design, and further developed a simple yet strong network backbone that not only scales well across *all* model sizes, but also transfers well to *multiple* core downstream tasks, surprisingly outperforming many other specifically designed methods.
>
> Arguably, a paper related to network backbone design may involve heavy engineering efforts. We emphasize that we have carefully discussed every design choice that leads to our final MOAT design, which may be clear in hindsight, but has never been explored by any of the existing works. We sincerely believe that our finding of MOAT could bring contributions to the community by establishing a *simple, strong and solid* baseline regarding how to effectively and seamlessly integrate MBConv and Transformer block for both upstream and downstream tasks.
>
> Finally, we would like to kindly ask the reviewer for the definition of “technical report” provided in any top-tier conference websites, and why a “technical report” is not suitable for the top-tier conferences. In our humble opinion, somes works [A, B, C, D, E], involving a reasonable amount of engineering efforts, could still benefit the community and advance the science, perhaps marginally but at a steady step.
>
> [A] Sun, Chen, et al. "Revisiting unreasonable effectiveness of data in deep learning era." ICCV 2017.
>
> [B] Bello, Irwan, et al. "Revisiting resnets: Improved training and scaling strategies." NeurIPS 2021 Spotlight.
>
> [C] Ghiasi, Golnaz, et al. "Simple copy-paste is a strong data augmentation method for instance segmentation." CVPR 2021 Oral.
>
> [D] Chen, Xinlei, Saining Xie, and Kaiming He. "An empirical study of training self-supervised vision transformers." ICCV 2021 Oral.
>
> [E] Dai, Zihang, et al. "Coatnet: Marrying convolution and attention for all data sizes." NeurIPS 2021.

---

### Author Response · Authors · 2022-11-28
**We are happy to engage in any author-reviewer discussion**

Dear Reviewers,

First of all, we would like to thank all of you for the constructive and valuable comments.
We understand that the author-reviewer discussion period has begun for a while, but we have not heard from any of you.
Please kindly read our responses along with other reviewer comments.
If you need more information or clarification, we are more than happy to provide and discuss with you.

Sincerely,

---

### Decision · Program_Chairs · 2023-01-20

**Decision:**

Accept: poster

**Justification For Why Not Higher Score:**

Some concerns around novelty remain

**Justification For Why Not Lower Score:**

Results still hold value to community.

**Metareview: Summary, Strengths And Weaknesses:**

Paper Summary:
Authors study a proposed simple concatenation of two network architectures, MBConv and Transformer, referring to it as a "MOAT block". Extensive experiments are performed on variations of this architecture, demonstrating improvements in performance across several vision tasks.

Review Summary:

Pros:
- Writing is good and easy to follow (GsU5,BAq9)
- Idea is simple and effective (GsU5,bFXs, W4v5,BAq9)
- Experiments are sufficient (GsU5,bFXs,W4v5)
- Open sourced code (W4v5)

Cons:
- Novelty is low. Simply joined two different architectures. (GsU5,W4v5) -- Authors do not sufficiently counter this critique, because it is true. However, the performance gain of this particular variation is still significant, and AC understands the amount of effort involved in rigorously testing new untried designs, simple as they may be. The results are still useful for the community.
- Some other relevant works are not mentioned (BAq9) -- Authors have added some more experiments.


AC Recommendation: Although there are concerns about novelty, reviews are unanimously leaning accept. AC appreciates comments about novelty, but the overall utility to the community given the improvements in performance is still high enough worthwhile for publication.

**Note From Pc:**

if the above contains the word "oral" or "spotlight" please see: "oral" presentation means -> notable-top-5% and "spotlight" means -> notable-top-25%. As stated in our emails, we are disassociating presentation type from AC recommendations